# E-cadherin mediates apical membrane initiation site localisation during *de novo* polarisation of epithelial cavities

Xuan Liang[1] (iD), Antonia Weberling[1] (iD), Chun Yuan Hii[1] (iD), Magdalena Zernicka-Goetz[1,2] (iD) & Clare E Buckley[1,*] (iD)

## Abstract

Individual cells within *de novo* polarising tubes and cavities must integrate their forming apical domains into a centralised apical membrane initiation site (AMIS). This is necessary to enable organised lumen formation within multi-cellular tissue. Despite the well-documented importance of cell division in localising the AMIS, we have found a division-independent mechanism of AMIS localisation that relies instead on Cadherin-mediated cell–cell adhesion. Our study of *de novo* polarising mouse embryonic stem cells (mESCs) cultured in 3D suggests that cell–cell adhesion localises apical proteins such as PAR-6 to a centralised AMIS. Unexpectedly, we also found that mESC clusters lacking functional E-cadherin still formed a lumen-like cavity in the absence of AMIS localisation but did so at a later stage of development via a "closure" mechanism, instead of via hollowing. This work suggests that there are two, interrelated mechanisms of apical polarity localisation: cell adhesion and cell division. Alignment of these mechanisms in space allows for redundancy in the system and ensures the development of a coherent epithelial structure within a growing organ.

**Keywords** AMIS; apical-basal polarity; cadherin; *de novo* polarisation; epithelial tube

**Subject Categories** Cell Adhesion, Polarity & Cytoskeleton

**The EMBO Journal (2022) 41: e111021**

See also: **G Herranz & F Martín-Belmonte** (December 2022)

## Introduction

Most organs in the body arise from tubes or cavities made from polarised epithelial cells. These cells have a strict apico-basal orientation; they align their apical ends along a centrally located lumen. Some tubes, such as the anterior neural tube in amniotes, arise via folding and closure of an already polarised epithelial tissue, through mechanisms such as actomyosin-mediated apical constriction (Nikolopoulou *et al*, 2017). However, many tubes and cavities, such as the posterior neural tube, mammary acini, kidney tubules and mammalian epiblast, arise via apical-basal polarisation within the centre of an initially solid tissue. The mechanisms by which such "*de novo*" polarisation is coordinated within dynamically growing tissue have been the focus of a significant body of research from several different models and have relevance both for understanding polarity-associated diseases and for directing organ bioengineering approaches.

Although the exact mechanisms are still under debate and may differ in different epithelia (Buckley & Johnston, 2022), Laminin, Integrin β1 and RAC1 signalling from the extracellular matrix (ECM) is now well established to be necessary for directing the overall apico-basal axis of polarisation of internally polarising tubes (Yu *et al*, 2004; Akhtar & Streuli, 2013; Buckley *et al*, 2013; Bedzhov & Zernicka-Goetz, 2014; Bryant *et al*, 2014; Molè *et al*, 2021). What is less clear is how the precise localisation of the apical membrane initiation site (AMIS) is directed at the single-cell level and how this is coordinated between neighbouring cells. The AMIS is a transient structure, marked by the scaffolding protein Partitioning-defective-3 (PAR-3) and tight junctional components such as Zonula occludens-1 (ZO-1), that defines where apically targeted proteins will fuse with the membrane, therefore determining where the lumen will arise (Bryant *et al*, 2010; Blasky *et al*, 2015). It is important that the subcellular localisation of the AMIS is coordinated between cells during morphogenesis to enable organised lumen formation.

The current literature suggests that cell division plays an important role in AMIS localisation. In particular, the post-mitotic mid-body has been shown to anchor apically directed proteins (Schlüter *et al*, 2009; Li *et al*, 2014; Wang *et al*, 2014; Luján *et al*, 2016; Rathbun *et al*, 2020). However, studies within the zebrafish neural rod showed that, whilst misorientation of cell division results in disruption of the apical plane at a tissue level, these phenotypes can be rescued by inhibiting cell division (Ciruna *et al*, 2006; Tawk *et al*, 2007; Quesada-Hernandez *et al*, 2010; Žigman *et al*, 2011).

---

1  Department of Physiology, Development and Neuroscience, University of Cambridge, Cambridge, UK
2  Division of Biology and Biological Engineering, California Institute of Technology, Pasadena, CA, USA
   *Corresponding author. Tel: +44 (0)1223 333766; E-mail: ceb85@cam.ac.uk

We also previously demonstrated that individual zebrafish neuroepithelial cells were able to recognise the future midline of the neural primordium and organise their intracellular structure around this location in advance and independently of cell division. This resulted in the initiation of an apical surface at whichever point the cells intersect the middle of the developing tissue, even if this is part way along a cell length (Buckley *et al*, 2013). This suggests that, whilst cell division is undoubtably a dominant mechanism, there must be another overlying mechanism driving AMIS localisation during *de novo* polarisation. The earliest indication of midline positioning in the zebrafish neural rod is the central accumulation of the junctional scaffolding protein Pard3 (PAR-3) and the adhesion protein N-cadherin (Buckley *et al*, 2013; Symonds & Buckley *et al*, 2020). This led us to hypothesise that cell–cell adhesions could direct the site for AMIS localisation during *de novo* polarisation. In line with this hypothesis, β-catenin-mediated maturation of N-cadherin was found to be necessary for the recruitment of the PAR apical complex protein atypical protein kinase C (aPKC) in the chick neural tube (Herrera *et al*, 2021). Opposing localisations of ECM and Cadherin proteins were also found to be sufficient to specify the apical-basal axis of hepatocytes in culture (Zhang *et al*, 2020).

To test the role of cell–cell adhesions in AMIS localisation, we turned to mouse embryo stem cell (mESC) culture in Matrigel, which has been used as an *in vitro* model for the *de novo* polarisation of the mouse epiblast (Bedzhov & Zernicka-Goetz, 2014; Shahbazi *et al*, 2017; Kim *et al*, 2021; Molè *et al*, 2021). This allowed us to study the initiation of apico-basal polarity of embryonic cells alongside the first cell–cell contacts between isolated cells and small cell clusters. It also allowed us to determine within a mammalian model whether division-independent polarisation is a conserved feature of *de novo* polarising structures. Unlike vertebrate epithelial cell culture models such as Madin–Darby canine kidney (MDCK) cells, which initiate lumenogenesis as early as the 2-cell stage when cultured in Matrigel (Bryant *et al*, 2010; Blasky *et al*, 2015), mESC cells in Matrigel only form lumens at the multi-cellular stage after 48–72 h in culture, coinciding with an exit in pluripotency (Bedzhov & Zernicka-Goetz, 2014; Shahbazi *et al*, 2017). This results in a relatively clear separation of the stages of *de novo* polarisation (Fig 1A). Previous literature suggests that the AMIS is formed at the 2-cell stage, around 24–36 h after culture in Matrigel, as denoted by membrane-localised PAR-3 and ZO-1 and sub-apical localisation of apical proteins such as Podocalyxin (PODXYL) (Shahbazi *et al*, 2017). The pre-apical patch (PAP) stage is formed after 36–48 h in culture, as denoted by the fusion of apical proteins such as PODXYL, PAR-6 and aPKC to the apical membrane and the displacement of junctional proteins PAR-3, ZO-1 and E-cadherin to the apico-lateral junctions (Shahbazi *et al*, 2017; Kim *et al*, 2021), following which lumenogenesis is initiated after 48–72 h in culture.

To determine the role of cell division and of cell adhesion in mESC AMIS localisation, we analysed mESC cells cultured in Matrigel at the AMIS stage with and without cell division in wild-type and E-cadherin knockout cell lines. We then further analysed polarisation and lumenogenesis in the absence of E-cadherin. Our results suggest that there is a division-independent mechanism of AMIS localisation that relies instead on E-cadherin-mediated cell–cell adhesions.

# Results

## Cell division is dispensable for AMIS localisation

First, we tested whether cell division was necessary for AMIS localisation in mESC rosettes. We cultured naïve, unpolarised mESCs (ES-E14 cells) in 2D on gelatin with 2i/LIF and then treated them with mitomycin C to block cell division. We then isolated single cells and seeded them into Matrigel without 2i/LIF, in N2B27 differentiation medium (Fig 1B). Cell divisions were efficiently blocked during the first 24-h post-seeding, during which time individual cells contacted each other and formed cell clusters in the absence of cell division (Movie EV1; Fig EV1A and B).

To assess AMIS localisation, we carried out immunofluorescence (IF) for PAR-3 and ZO-1 at 24-h post-seeding. As previously published (Shahbazi *et al*, 2017), in addition to several puncta at the cell peripheries, both PAR-3 and ZO-1 localised to the membrane at the centre of control cell–cell contacts, marking the AMIS in the majority of cell clusters (Fig 1C(i) and H(i)). Interestingly, division-blocked cells also localised PAR-3 and ZO-1 to the central membrane (Fig 1C(ii) and H(ii), quantified in Fig 1D–G and I). In both control and division-blocked cell clusters, there was a small proportion that had not yet fully localised the AMIS at the 24-h stage (Fig 1D and I), where PAR-3 was either not localised (Appendix Fig S1A) or was only weakly present at cell–cell interfaces (Appendix Fig S1B). E-cadherin was upregulated along the whole length of the cell–cell interfaces in both dividing and non-dividing cell clusters, with a higher level of E-cadherin at the cell–cell interface relative to the cell-matrix interface (Figs 1C and EV1C). To quantify the subcellular localisation of PAR-3 in each cell cluster, we carried out intensity profiles across the cell–cell interface of 2-cell doublets (Fig 1E(a) and F) and calculated the ratio between centralised and surrounding non-centralised PAR-3 in multi-cellular clusters (Fig 1E(b) and G). This confirmed that PAR-3 localised to a small central area at the cell–cell interface in both control and division-blocked 2-cell doublets and multi-cellular clusters. Golgi apparatus and centrosomes were also localised to the centre of cell–cell contacts both in dividing and non-dividing conditions (Fig 1H–K; Appendix Fig S1C), confirming that mESCs were polarised in the absence of cell division. mESC clusters also centrally localised PAR-3, ZO-1 and polarised the Golgi apparatus when cell division was blocked using an alternative compound, aphidicolin (Fig EV1D–H).

Together, these results demonstrate that cell division is dispensable for *de novo* AMIS localisation in polarising mESCs.

## Cell–cell contact directs PAR-6 localisation

To understand the dynamics of apical protein polarisation in the absence of cell division, we generated a mESC stable cell line expressing mCherry-PAR-6B and imaged cells live. In line with the previous characterisation of PAR-6 by IF (Shahbazi *et al*, 2017; Kim *et al*, 2021) in control dividing cells, mCherry-PAR-6B localised to the apical membrane by the PAP stage at 48 h and to the luminal apical membrane from 72 h (Fig 2A; Appendix Fig S2A and B). In addition, the transgene allowed us to better visualise PAR-6B puncta at earlier 24-h AMIS stages of development. At this stage, mCherry-PAR-6B was localised sub-apically, polarised towards the central region of cell–cell contact (Fig 2A; Appendix Fig S2B). Some

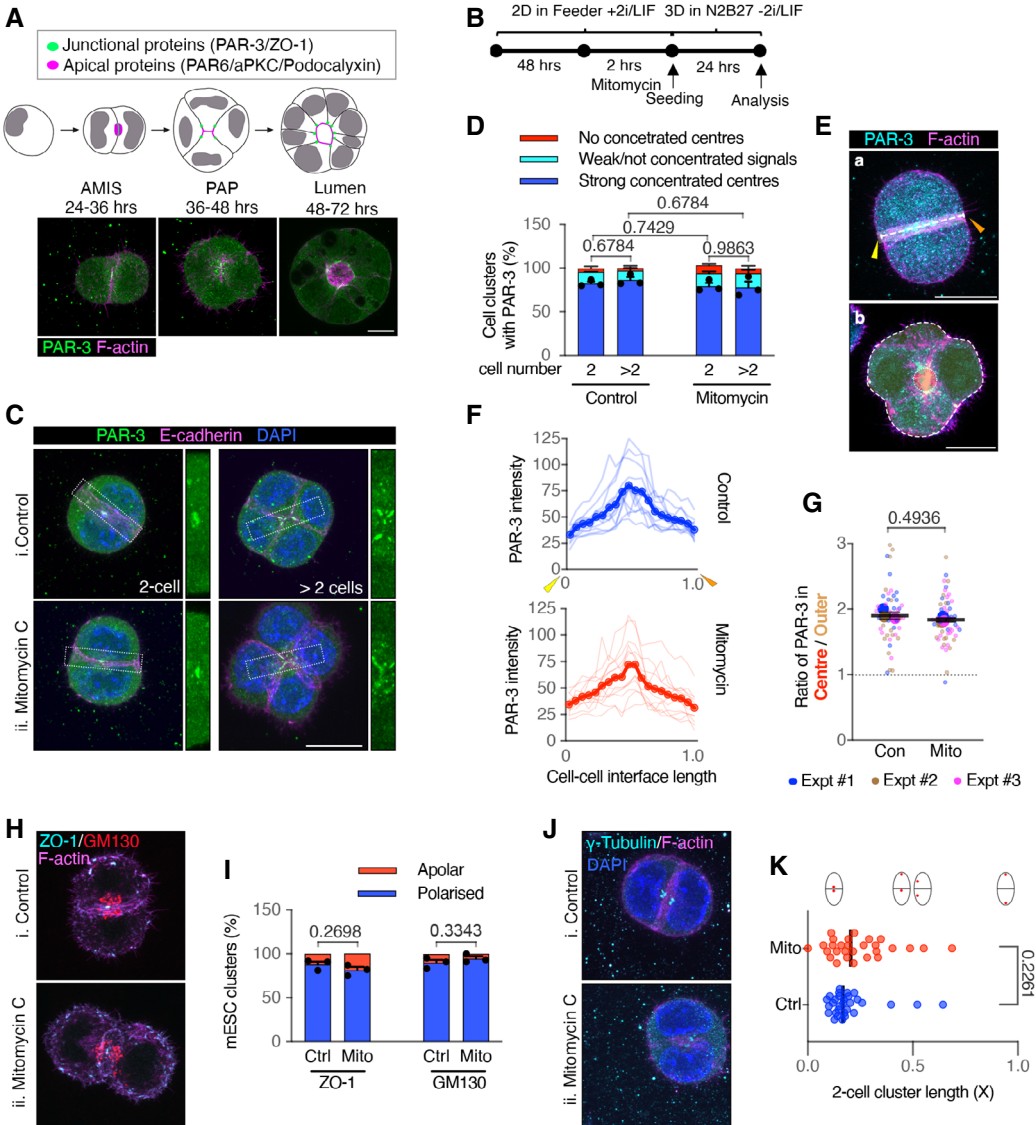

**Figure 1. Cell division is dispensable for AMIS localisation in mESC 3D cultures in Matrigel.**

A Stages of polarisation and lumen formation in mESCs cultured in Matrigel.

B Timeline of experiment setups to assess AMIS formation.

C Immunofluorescence of PAR-3 and E-cadherin. PAR-3 localisation was concentrated at the centre of 2-cell mESC doublets and 3- or 4-cell mESC clusters, whilst E-cadherin was localised along the whole length of cell–cell interfaces in both control and mitomycin C-treated conditions.

D Quantification of the frequency of cell clusters with a strong polarised PAR-3 centre. See the representative mESC clusters without the strong polarised PAR-3 in Appendix Fig S1A and B.

E Illustrations of PAR-3 analysis in 2-cell (a) and >2-cell (b) clusters. The line-scan analysis was performed from the yellow arrow to the orange arrow at the cell–cell interface between two cells in (a). The average pixel intensity was analysed at central (red region, inner dotted line) and surrounding non-central regions (orange region, between the inner and outer dotted lines) in (b).

F Line-scan profiles of PAR-3 at the cell–cell interface of 2-cell mESC doublets. Line-scans were sectioned and fitted to each 5% along the cell–cell interface length.

G Ratio of PAR-3 pixel intensity values at central and surrounding regions in >2-cell mESC clusters.

H, I Representative images of centralised ZO-1 puncta and polarised Golgi apparatus, labelled by GM130 (H) and quantification of the frequency of cell doublets with a polarised ZO-1 or Golgi apparatus centre (I). Also, see the spilt channels in Appendix Fig S1C.

J, K Representative images of polarised centrosomes (J), labelled with γ-tubulin and distance between centrosomes (K) normalised to the length of the long axis of the doublets in 2-cell clusters.

Data information: Data are presented as means ± SEM in (D) and (I); individual line-scans and mean-valued line-scans in (F); individual cell cluster values (small dots), mean experimental values (large dots) and means of three experiments (bars) in (G); individual values in dots and median values in bars in (K). n = 3 experiments in (D) and (I), 15–32 cell clusters were analysed for each column in every experiment; 15 cell doublets for each condition from one experiment in (F); 15–22 cell clusters for each condition in every experiment in (G); 35 doublets for each condition in (K). Two-way ANOVA analysis in (D); Student's *t*-test analysis in (G), (I), (K). *P*-values were listed in the graphs. All scale bars: 10 μm.

Source data are available online for this figure.

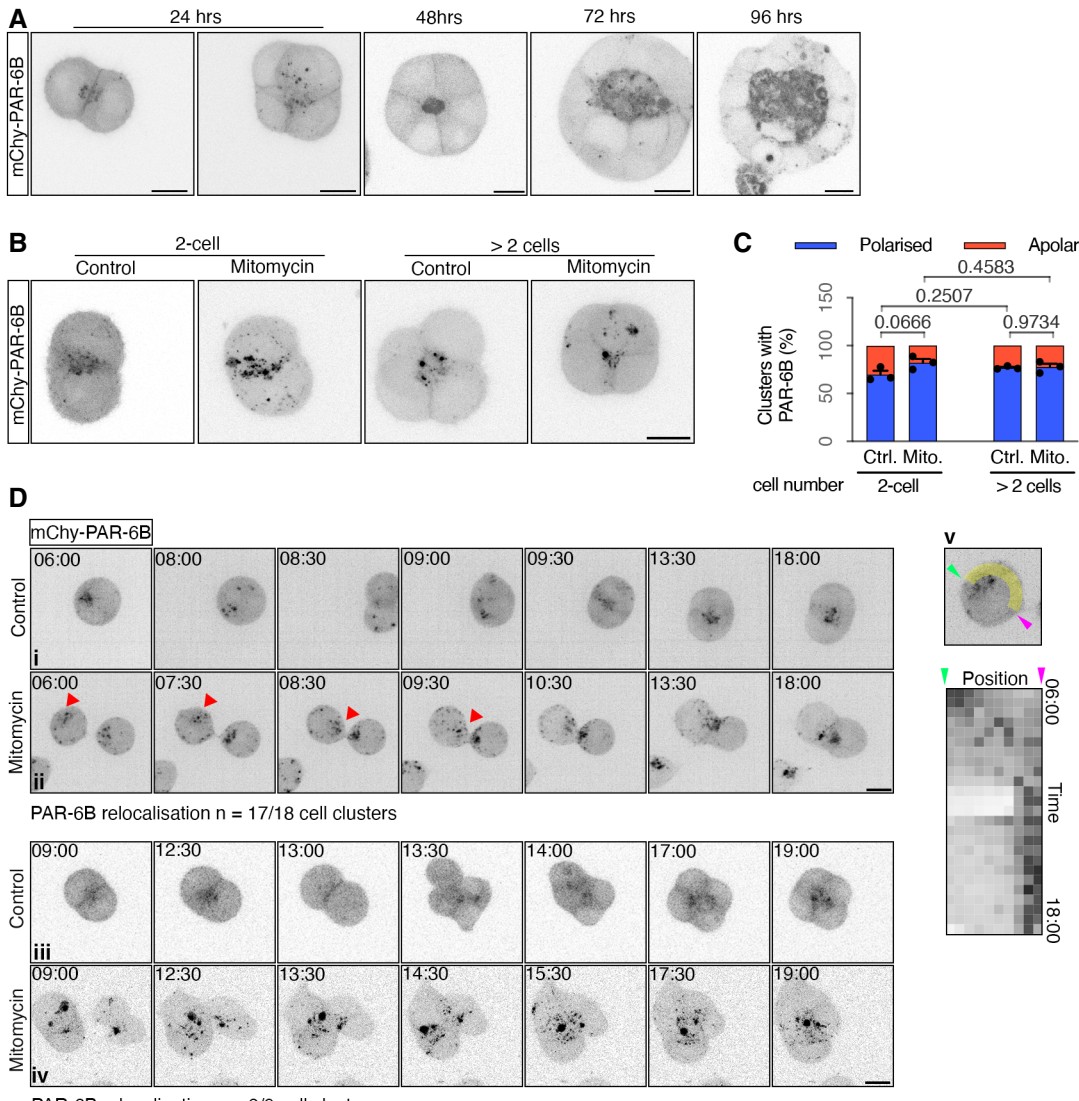

**Figure 2. Polarised PAR6B in mESC 3D cultures.**

A    Representative images of mCherry-PAR-6B mESC live cells cultured from 24–96 h in Matrigel. See the bright-field images in Appendix Fig S2A.

B, C    Representative images (B) and quantification of the frequency of cell clusters with polarised mCherry-PAR-6B (C) in control and mitomycin division-blocked mESC live cells after 24 h in Matrigel.

D    Movie stills of mCherry-PAR-6B in control and mitomycin C-treated mESCs cultured in Matrigel (also Movie EV2). Control cells divided (i) and two mitomycin-treated cells touched (ii) to form 2-cell doublets. Control cells divided twice (iii) and two mitomycin-treated cell clusters touched (iv) to form 4-cell clusters. (v), Kymograph of mCherry-PAR6B in the arrow-head cell in (ii) along the path between the arrows; each pixel is the fluorescence values averaged over 0.2 μm sections. See more examples in Appendix Fig S2D and E.

Data information: Data in (C) are presented as means ± SEM. n = 3 experiments. At least 20 clusters were analysed for each column in every experiment. Two-way ANOVA analysis; P-values were listed in the graphs. All scale bars: 10 μm.

Source data are available online for this figure.

mCherry-PAR-6B puncta appeared to be associated with the Golgi network. However, a large proportion of mCherry-PAR-6B was in the cytoplasm sub-apically (Appendix Fig S2C), suggesting that PAR-6B puncta were in the process of being delivered to the apical membrane at 24 h. A similar polarised distribution of PAR-6B was observed in both control and division-blocked cells (Fig 2B and C; Appendix Fig S2C), demonstrating that cell division is dispensable for apical protein polarisation.

We next assessed the dynamics of PAR-6B polarisation. In both control and division-blocked cells, non-cortical mCherry-PAR-6B puncta were visible at the single-cell stage. In control cells, these puncta relocalised to the abscission plane, following cell division (Fig 2D; Appendix Fig S2D; Movie EV2). In division-blocked cells, PAR-6B puncta dynamically relocalised to newly forming cell–cell contacts, eventually forming cell–cell clusters with centrally localised PAR-6B (Fig 2D; Appendix Fig S2E; Movie EV2).

These results suggest that cell–cell contact directs PAR-6B localisation at the central AMIS, independent of cell division.

### E-cadherin adhesions are necessary for AMIS localisation

The above results suggest that there is a division-independent mechanism of AMIS localisation that relies instead on cell–cell adhesions. Since E-cadherin is the predominant adhesion molecule in nonneural epithelia, we hypothesised that it might be important in AMIS localisation. To achieve a full removal of E-cadherin, we employed an *E-cadherin* knockout (*Cdh1* KO) mESC line (Appendix Fig S3A; Larue *et al*, 1996).

To assess AMIS localisation, we again carried out IF for PAR-3 and ZO-1 at 24-h post-seeding. As seen earlier (Fig 1), both PAR-3 and ZO-1 localised to the central region of cell–cell contact within wild-type (W4 cells) doublets/clusters with and without division. However, PAR-3 and ZO-1 localisation was strongly inhibited in the absence of E-cadherin (Fig 3A–D, G, and H). RNAi knock-down (KD) of E-cadherin in ES-E14 mESCs showed similar results to the *Cdh1* KO mESCs: PAR-3 at the central region of E-cadherin KD two-cell clusters was significantly reduced (Appendix Fig S3B and C).

To investigate AMIS localisation at a single-cell level, we co-cultured division-blocked wild-type (ES-E14) and *Cdh1* KO cells and analysed chimeric mESC doublets, comprising one wild-type and one *Cdh1* KO cell. Whilst homogenous control doublets localised PAR-3 to the central region of the cell–cell interface, heterogeneous chimeric doublets did not localise PAR-3 centrally (Fig 3E and F). The same result was seen in E-cadherin RNAi chimeric doublets (Appendix Fig S3B(b)). Golgi and centrosome localisation towards the cell–cell interface suggested that the overall axis of polarity was maintained, even in the absence of both cell division and E-cadherin (Fig 3G–I; Appendix Fig S3D). These results demonstrated that E-cadherin is necessary for AMIS localisation.

Since E-cadherin is localised along the whole cell–cell interface but PAR-3 and ZO-1 localise at central cell–cell interfaces, we next used fluorescence recovery after photobleaching (FRAP) to compare the stability of E-cadherin protein at central and side regions in 2-cell mESC clusters (Fig EV2A and B, regions illustrated in Fig EV2C). E-cadherin-eGFP (Fig EV2B) and E-cadherin immunofluorescence (Fig EV2C and D) levels were the same at these two regions. However, FRAP of E-cadherin-eGFP showed that the mobile fraction of E-cadherin-eGFP was lower in the central region than the side regions (Fig 3J and K). Therefore, E-cadherin junctions are more stable at the centre-most region of the cell–cell interface, which may provide at least a partial explanation for why AMIS localisation occurs precisely at this region.

It has previously been demonstrated that a reduction in E-cadherin can slow pluripotency exit (Soncin *et al*, 2009). However, pluripotency exit was previously shown not to alter AMIS formation (Shahbazi *et al*, 2017). In support of these results, we also found that cells maintained in the pluripotent state when cultured in Feeder Cell medium provided with 2i/LIF still localised the AMIS, with and without cell division (Fig EV3A–C). However, in line with our results showing lack of AMIS localisation in *Cdh1* KO cells cultured in the absence of 2i/LIF (Fig 3), cells cultured in the presence of 2i/LIF also could not localise an AMIS in the absence of E-cadherin (Fig EV3A–C). Despite this result, we wanted to check whether the stage of pluripotency exit differed between WT and *Cdh1* KO cells in our experiments since this might indicate a different speed of maturation. We, therefore, carried out IF for Orthodenticle Homeobox 2 (OTX2) protein, which is necessary for pluripotency exit, and the pluripotency marker protein Nanog. Although, as expected, the overall level of nuclear OTX2 increased and Nanog decreased over the 24-h course of development, we found no significant difference in post-mitotic levels of OTX2 and Nanog between WT and *Cdh1* KO cells (Fig EV3D). This result suggests that there was no difference in the stage of pluripotency exit in the cell clusters that we analysed during this study, and this is therefore unlikely to play a role in the lack of AMIS localisation seen in *Cdh1* KO cells.

These results demonstrate that E-cadherin adhesions between cells are necessary for AMIS localisation but not for the overall axis of polarity. They also demonstrate that ECM in the absence of E-cadherin is insufficient for AMIS localisation.

**Figure 3. E-cadherin junctions are important for polarisation.**

A, B   Immunofluorescence of PAR-3 (A) and proportions of mESC clusters with a strong PAR-3 centre (B) in wild-type (W4) and E-cadherin knockout (*Cdh1* KO) mESCs at 24 h in Matrigel. See E-cadherin in Appendix Fig S3A.

C   Line-scan profiles of PAR-3 at the cell–cell interface in wild-type control, mitomycin C-treated and *Cdh1* KO control, mitomycin C-treated 2-cell clusters.

D   Ratio of PAR-3 pixel intensity values at central and surrounding regions in >2-cell mESC clusters.

E, F   Representative images of PAR-3 immunofluorescence (E) and line-scan profiles of PAR-3 at the cell–cell interface (F) in WT homogeneous (ES-E14) or WT/*Cdh1* KO chimeric mESC 2-cell doublets. *, WT mESCs.

G, H   Representative images of ZO-1 puncta and Golgi apparatus (G) and proportions of mESC doublets with central ZO-1 puncta or polarised Golgi apparatus (H) in WT and *Cdh1* KO mESC doublets.

I   Distance between centrosomes in cell doublets. The distance was normalised to the length of the long axis of the doubles. See the representative images in Appendix Fig S3D.

J   Fluorescence recovery after photobleaching (FRAP) of E-cadherin-eGFP at the centre-most or side 2 μm regions of division-blocked mESC doublets' cell–cell interfaces. See methods in Fig EV2C.

K   Mobile fraction of E-cadherin-eGFP calculated from the plotting of (J).

Data information: Data are presented as means ± SEM in (B), (H); individual and mean-valued line-scans in (C) and (F); individual cell cluster values (small dots), mean experimental values (large dots) and means of three experiments (bars) ± SEM in (D); individual values in dots and median values in bars in (I); exponential association fitting curves ± SD in (J); means ± SD in (K). n = 3 experiments in (B) and (H), 17–50 cell clusters were analysed for each column in every experiment; 15 doublets for each condition in (C); 17–30 cell clusters were analysed for each condition in every experiment in (D); 15 doublets for each condition in (F); 35–40 doublets in each condition in (I); 15–18 doublets for each condition in (J) and (K). Two-way ANOVA analysis in (B), (D), (I) and (K); Student's *t*-test analysis in (H); *P*-values were listed in the graphs. All scale bars: 10 μm.

Source data are available online for this figure.

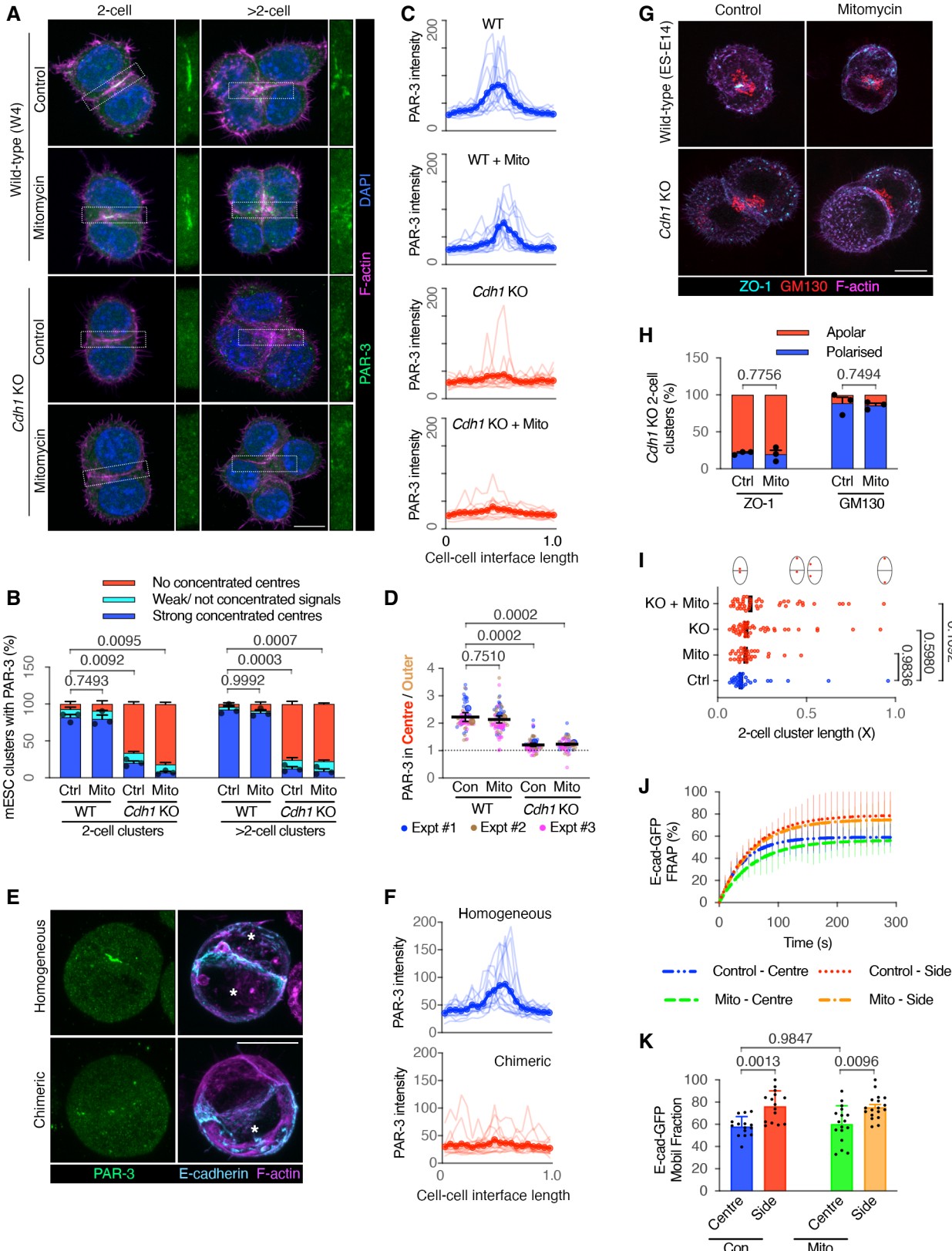

Figure 3.

## Adhesion molecules P-cadherin, JAM-A and Nectin-2 are not necessary for AMIS localisation

E-cadherin is not the only form of adhesion molecule that is expressed at cell–cell contacts. A complex network of interactions between the PAR-complex, adhesion molecules, MAGUK scaffolding proteins and the actin cytoskeleton is responsible for building cell–cell junctions (Buckley & Johnston, 2022). Of relevance to this study, PAR-3 has been found to directly bind to transmembrane Junctional Adhesion Molecules (JAMs) and Nectin proteins in mammals (Ebnet *et al*, 2001; Itoh *et al*, 2001; Takekuni *et al*, 2003). In the mammalian embryo, JAM-A and Nectin-2 adhesion molecules are expressed between inner cell mass cells in the mouse blastocyst (Thomas *et al*, 2004). We found that JAM-A and Nectin-2, as well as P-cadherin, were expressed at cell–cell contacts in 2D cultured mESCs (Fig EV4A).

We, therefore, carried out IF of 2-cell mESC clusters cultured in Matrigel for 24 h to determine the localisation of P-cadherin, JAM-A and Nectin-2 at the AMIS stage. Whilst the majority of 2-cell and 4-cell clusters formed a polarised PAR-3 centre, P-cadherin was uniformly expressed along cell–cell interfaces (Figs 4A and EV4B). JAM-A was uniformly expressed along cell–cell interfaces at the 2-cell stage (Fig 4D) and concentrated towards the centre of 4-cell mESC clusters (Fig EV4C). At the 2-cell stage, Nectin-2 was expressed along the whole cell–cell interface with a sight concentration towards the central regions where PAR-3 was localised (Fig 4G). At the 4-cell stage, Nectin-2 was concentrated towards the centre of the clusters (Fig EV4D). These results suggest that PAR-3 localises at the AMIS before JAM-A or Nectin-2.

Next, we used siRNA KD of protein function in dividing and division-blocked cells to test whether P-cadherin, JAM-A or Nectin-2 proteins were necessary for AMIS localisation. However, following siRNA for each of these proteins, PAR-3 was still polarised to the centre of cell–cell contacts (Figs 4A–I and EV4B–D). Compared with the loss of PAR-3 polarisation upon E-cadherin KO or KD (Fig 3, Appendix Fig S3), the results demonstrate that P-cadherin, JAM-A and Nectin-2 are not necessary for AMIS localisation. Indeed, when the centralised PAR-3 localisation was lost in the E-cadherin KO mESC 2-cell clusters (Fig 3A–C), P-cadherin, JAM-A and Nectin-2 were still expressed at the cell–cell interface between the mESCs (Fig EV4E–G). This suggests that E-cadherin-based adhesions might be specifically responsible for mediating AMIS localisation.

## E-cadherin adhesions are sufficient to initiate AMIS localisation, independent of ECM signalling and cell division

As discussed, ECM-mediated signalling plays an important role in orienting the axis of polarisation within *de novo* polarising systems. Recently, the apico-basal axis of cultured mature hepatocytes was established by a combination of ECM signalling and immobilised E-cadherin (Zhang *et al*, 2020). However, PAR-3 has also recently been shown to polarise in mESCs lacking functional Integrin-β1 or cultured in agarose in the absence of ECM proteins (Molè *et al*, 2021). Our current study shows that the AMIS can localise in the absence of cell division but not in the absence of E-cadherin. We, therefore, wanted to explore the relative roles of ECM, cell division and E-cadherin in AMIS localisation.

We first eliminated the influence of ECM by culturing division-blocked mESCs (ES-E14) in 0.5% agarose and carried out IF for PAR-3 after 30 h in culture. These cells were still able to polarise PAR-3, even in the absence of both cell division and ECM proteins (Fig 5A and B). However, in line with our earlier results (Fig 3), PAR-3 localisation was strongly inhibited in *Cdh1* KO cells (Fig 5A and B). These results suggest that AMIS localisation occurs independently of both ECM signalling and of cell division, relying instead on E-cadherin.

To further test the sufficiency of E-cadherin adhesions in initiating AMIS localisation, we cultured individual division-blocked mESCs (ES-E14) onto either E-cadherin recombinant protein or Fibronectin precoated glass, then topped the cells with N2B27 medium, with or without 20% Matrigel and carried out IF for PAR-3 after 24 h in culture. Like results from hepatocytes (Zhang *et al*, 2020), cells plated on E-cadherin and topped with Matrigel localised PAR-3 to the centre of the cell-cadherin interface (Fig 5C–F). However, this central PAR-3 localisation was significantly reduced when cells were plated on fibronectin (Fig 5C–F). Interestingly, cells cultured on E-cadherin but in the absence of Matrigel still localised PAR-3 to the centre of the cell-cadherin interface (Fig 5C–F).

These results demonstrate that E-cadherin adhesions are both necessary and sufficient for initiating AMIS localisation, whilst ECM is not necessary or sufficient for AMIS localisation.

## E-cadherin is necessary for hollowing lumenogenesis

We next wanted to test the importance of E-cadherin-mediated AMIS localisation in lumenogenesis. We, therefore, cultured WT (ES-E14) and *Cdh1* KO mESCs and fixed them at the AMIS 24-h stage, PAP 48-h stage and lumen 72 and 96-h stage. We then carried out IF for PAR-3 and ZO-1 to label AMIS/apical-lateral junctions and PODXYL to label apical proteins. Whilst most WT cell clusters had a centralised apical domain or small lumen after 48 h in culture, very few *Cdh1* KO cell clusters had made a centralised apical domain by the 48-h PAP stage (Fig 6A–D). In line with our earlier findings at the 24-h AMIS stage (Fig 3), this provides further evidence that E-cadherin is necessary for centralised AMIS localisation. However, we noticed that a small percentage of *Cdh1* KO cell clusters at 48 h had formed an open "cup-shape," with apically localised PODXYL (Fig 6B) and apico-laterally localised junctional PAR-3 (e.g. Fig 6A(iii)) and ZO-1 (Fig EV5A). We termed these "open cavities" (Fig 6C). Surprisingly, by the 72-h lumen stage, approximately 75% of *Cdh1* KO cell clusters had formed polarised cavities, approximately 50% of which were open cavities and 50% were closed (Fig 6D and E). Over the course of 48–96 h in culture, the overall percentage of polarised cavities increased (Fig 6D) as did the proportion of these structures that were "closed" (Fig 6E). This suggested that these cavities might form via gradual "closure" of the tissue, rather than via hollowing. Both "open" and "closed" cavities were surrounded by polarised Golgi apparatus, demonstrating that the overall apico-basal axis of cells was intact. (Fig EV5B).

To further assess the morphogenetic mechanism by which cavities form in *Cdh1* KO cells, we generated WT (ES-E14) and *Cdh1* KO mESC lines labelled with LifeAct-mRuby (Appendix Fig S4). We visualised the process of lumenogenesis within mESCs cultured in Matrigel via live imaging (Fig 6F; Movies EV3 and EV4). This

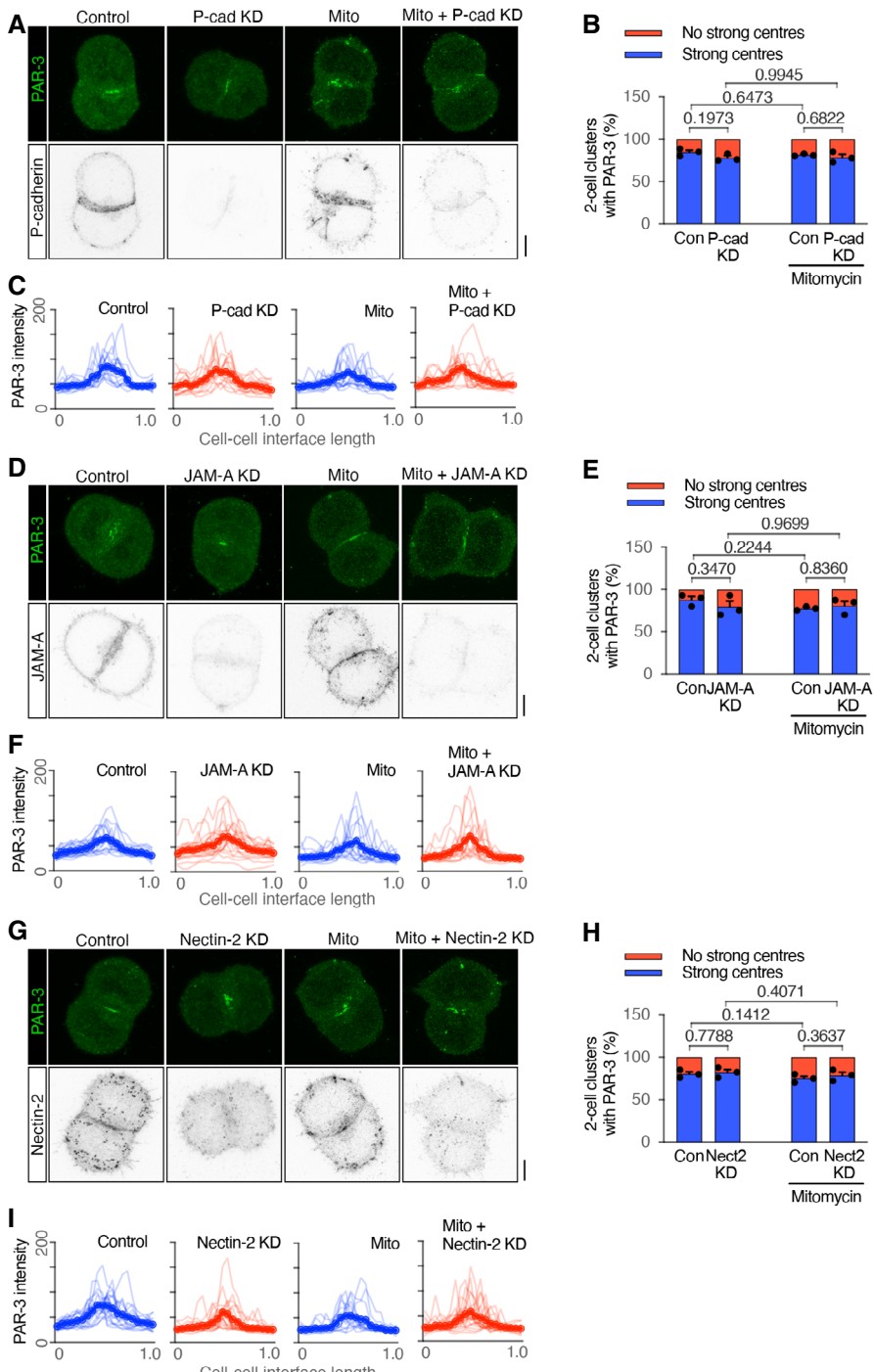

**Figure 4. Adhesion molecule P-cadherin, JAM-A and Nectin-2 do not aid PAR-3 in the AMIS.**

A–C   Immunofluorescence of PAR-3 and P-cadherin (A), proportions of mESC clusters with a positive PAR-3 centre (B) and line-scans of PAR-3 at the cell–cell interface (C) in control, P-cadherin knock-down by RNAi, mitomycin-treated and P-cadherin knock-down mitomycin-treated W4 mESC doublets cultured for 24 h in Matrigel.

D–F   Immunofluorescence of PAR-3 and JAM-A (D), proportions of mESC clusters with a positive PAR-3 centre (E) and line-scans of PAR-3 at the cell–cell interface (F) in control, JAM-A knock-down by RNAi, mitomycin-treated and JAM-A knock-down mitomycin-treated W4 mESC doublets cultured for 24 h in Matrigel.

G–I   Immunofluorescence of PAR-3 and Nectin-2 (G), proportions of mESC clusters with a positive PAR-3 centre (H) and line-scans of PAR-3 at the cell–cell interface (I) in control, Nectin-2 knock-down by RNAi, mitomycin-treated and Nectin-2 knock-down mitomycin-treated W4 mESC doublets cultured for 24 h in Matrigel.

Data information: Data are presented as means ± SEM in (B), (E), (H); individual and mean-valued line-scans in (C), (F), (I). *n* = 3 experiments in (B), (E), (H), 15–21 clusters were analysed for each column in every experiment; 15–21 line-scans in (C), (F), (I). Two-way ANOVA analysis in (B), (E), (H); *P*-values were listed in the graphs. All scale bars: 5 µm.

Source data are available online for this figure.

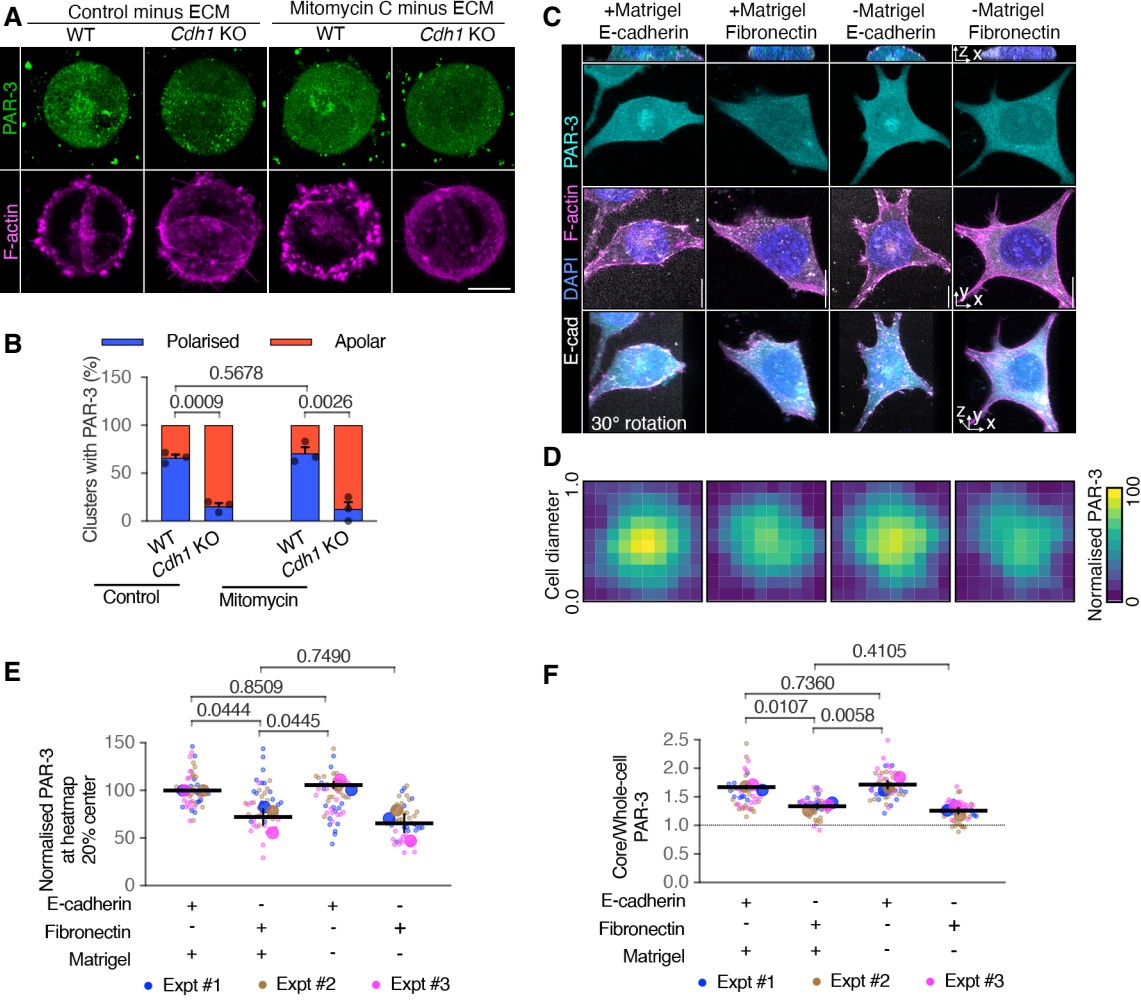

**Figure 5. Cell–ECM interactions in regulating AMIS seeding.**

A, B   Immunofluorescence of PAR-3 (A) and proportions of mESC doublets with polarised PAR-3 (B) in wild-type (ES-E14) and E-cadherin knockout (*Cdh1* KO) cells at 30 h in 0.5% agarose.

C   Immunofluorescence of PAR-3 in cell division-blocked mESCs cultured on E-cadherin or Fibronectin-coated glass topped with or without Matrigel for 24 h.

D   Heatmap of PAR-3 in cells from one experiment of (C). Squared frames were fitted to the main bodies of the cells. A pixel was the average value in each 10% along the width of the frames. The heatmaps were stacks of 15 cells in one experiment.

E   PAR-3 levels at the central 20% of the heatmaps from three experiments of (C).

F   Ratios between PAR-3 in the 2.5 μm diameter central core and whole cell surface from three experiments of (C).

Data information: Data are presented as means ± SEM in (B); individual cell values (small dots), mean experimental values (large dots) and means of experiments ± SEM (bars) in (E) and (F). *n* = 3 experiments in (B), (E), (F); at least 15 clusters were analysed for each column in every experiment; 15–20 cells for each column in every experiment of (E) and (F). Two-way ANOVA analysis in (B), (E), (F); *P*-values were listed in the graphs. All scale bars: 5 μm.

Source data are available online for this figure.

confirmed that, whilst the WT cell clusters made a central lumen (8/8 movies on day 2) and then expanded this already central lumen (8/8 movies on day 3), *cdh1* KO cell clusters first generated an open cup-shape cavity (3/3 movies on day 2), which then gradually closed, eventually generating a centralised lumen-like structure without hollowing at a later stage of development (3/3 movies on day 3).

These results demonstrate that, in the absence of E-cadherin-mediated AMIS localisation, cell clusters do not hollow but instead generate lumen-like cavities via a closure mechanism. Our results also demonstrate that E-cadherin and centralised AMIS localisation

are not required for apical membrane formation. In the absence of E-cadherin, an apical surface is still formed but this occurs later in development so appears less efficient.

# Discussion

### Epithelial cells can polarise *de novo* in the absence of cell division

Both AMIS-associated proteins PAR-3/ZO-1 and apical polarity protein PAR-6B localised similarly in WT and division-blocked

mESCs (Figs 1, 2, and 7A). This finding supports our previously published zebrafish neuroepithelial cell *in vivo* analysis, which demonstrated the division-independent localisation of Pard3 (PAR-3) and ZO-1 at the neural rod primordial midline (Buckley *et al*, 2013). Together, this demonstrates that although division is

an important contributor to AMIS formation, a division-independent mechanism of *de novo* polarisation and AMIS localisation can occur in both *in vivo* and *in vitro* conditions. Whilst disorganised lumen formation can also occur in the absence of division in the zebrafish neural rod (Buckley *et al*, 2013), this was not possible to test within

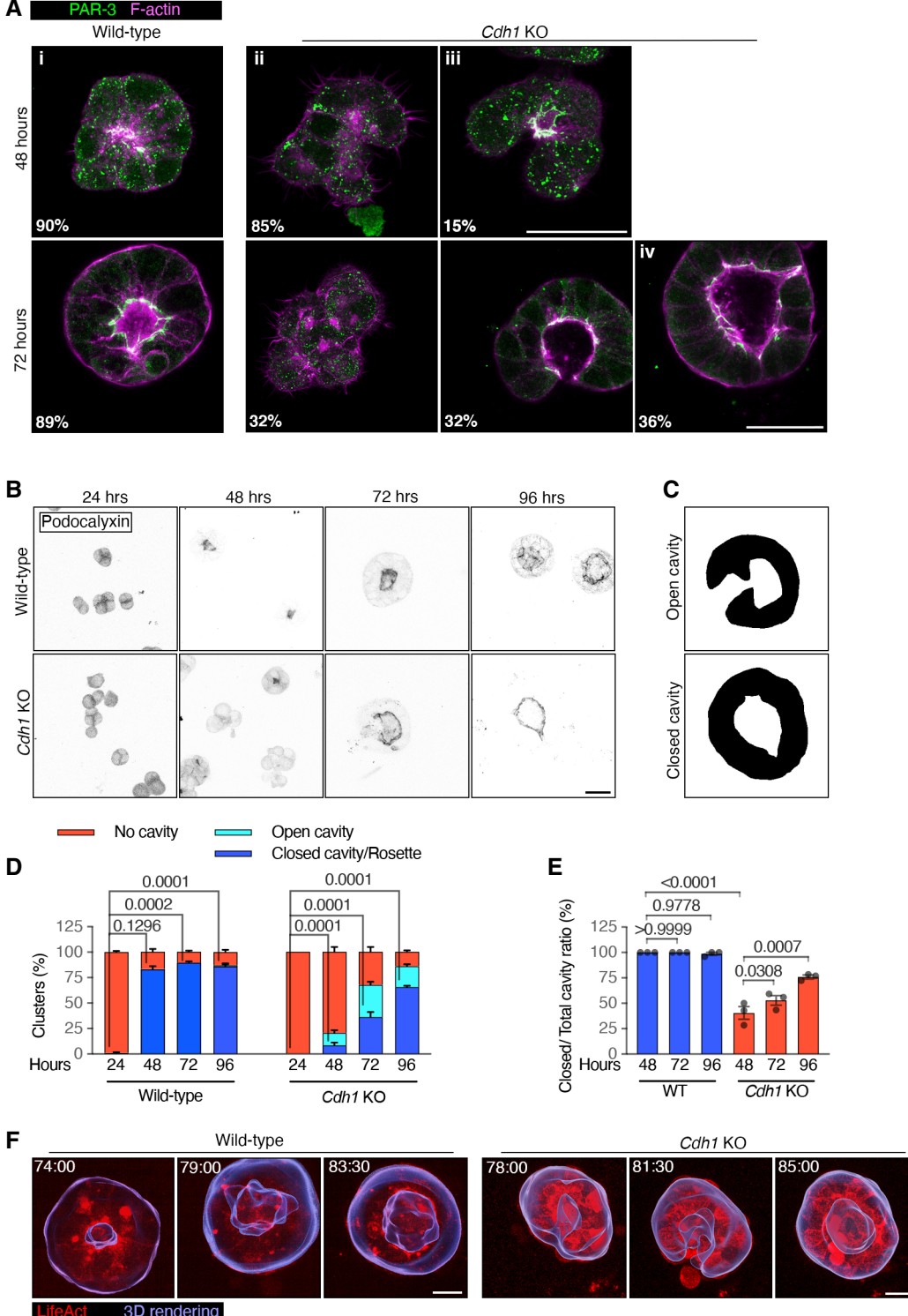

Figure 6.

◀

**Figure 6. Lumenogenesis in wild-type and E-cadherin knockout mESC cultures.**

A PAR-3 immunofluorescence in wild-type (ES-E14) and E-cadherin knockout (*Cdh1* KO) mESCs cultured in Matrigel for 48 and 72 h. At 48 h, most WT cell clusters had formed a polarised rosette or small lumen (i). However, most Cdh1 KO cells had not formed a central AMIS (ii). A small percentage of Cdh1 KO cells had formed an open "cup-shape," with apically localised PAR-3 (iii). By 72 h, a significant proportion of Cdh1 KO cells had formed polarised "cavity-like" structures, about half of which were configured in an open "cup-shape" and half as closed "lumen-like" structures (iv).

B Podocalyxin immunofluorescence in WT and Cdh1 KO mESCs cultured in Matrigel from 1–4 days. See Fig EV5A for ZO-1 staining.

C Examples of masked *Cdh1* KO cell cluster surfaces from "open" and "closed" cavities. The cavities were categorised based on Podocalyxin signals.

D Percentage of cell clusters with different cavities relative to total cell clusters at different time points. The analysis was compared between the closed/rosette category among the conditions.

E Percentage of cell clusters with closed cavities relative to total cell clusters with cavities calculated from (D).

F Movie stills of LifeAct-mRuby labelled cell clusters. The images are whole cell cluster z-projections, overlaid with 3D rendering of the cluster surfaces to show the forming lumens. See Movie EV3 for z-stack movies and Movie EV4 for 3D rotations.

Data information: Data are presented as means ± SEM in (D) and (E). *n* = 20 (48 h WT & *Cdh1* KO), 30 (72 h WT) and 32 (72 h *Cdh1* KO) images in (A); 3 experiments in (D) and (E), at least 25 clusters were analysed for each column in every experiment. Two-way ANOVA analysis in (D) and (E); *P*-values were listed in the graphs. All scale bars: 25 μm.

Source data are available online for this figure.

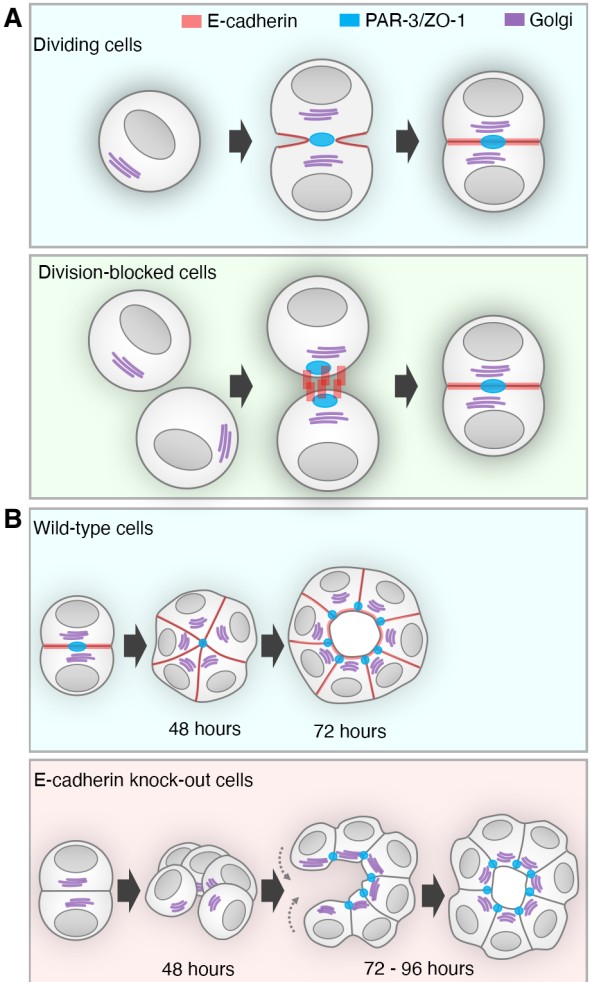

**Figure 7. Summary of findings.**

A *De novo* polarisation and AMIS formation in dividing and division-blocked mESCs cultured in Matrigel.

B Lumenogenesis in wide-type and E-cadherin knockout mESCs cultured in Matrigel.

the mESC culture model since mitomycin-treated cell clusters did not survive beyond 30 h in culture.

**E-cadherin-based cell–cell contacts are necessary and sufficient to initiate AMIS localisation**

AMIS localisation in *Cdh1* KO cell clusters is strongly inhibited (Fig 3), and individual mESC cells can localise their AMIS to the central region of the cell-cell interface independently from ECM signalling (Fig 5). Together, this demonstrates that the formation of E-cadherin-based cell–cell contacts is both necessary and sufficient for initiating AMIS localisation and that ECM is insufficient to direct AMIS localisation in the absence of E-cadherin. Our results, therefore, suggest that Cadherin-based cell–cell adhesion provides the spatial cue required for AMIS localisation during *de novo* polarisation. This in turn localises apical proteins such as PAR-6B to a centralised region of cell–cell contact (Fig 2), determining where the lumen will arise.

Whilst we demonstrate that AMIS localisation can occur independently from cell division, the importance of abscission and midbody formation in apical protein targeting has been robustly demonstrated and the molecules involved are now starting to emerge (Schlüter *et al*, 2009; Li *et al*, 2014; Wang *et al*, 2014, 2021; Klinkert *et al*, 2016; Luján *et al*, 2016; Mangan *et al*, 2016; Rathbun *et al*, 2020). Rather than acting as the initial symmetry-breaking step in AMIS localisation, we suggest that tethering of apically directed proteins to the midbody might instead act to transiently align cell division, cell adhesion and the forming apical domain, therefore enabling an organised structure to be generated in the presence of dynamic cell movement and tissue growth (Buckley & Johnston, 2022). The localisation of scaffolding and tight junction-associated proteins such as PAR-3 and ZO-1 at the AMIS might aid in this alignment. For example, Cingulin is a tight junctional protein that has been shown to bind both to the midbody and to FIP5, which is important for the apical targeting of vesicles containing apical proteins (Mangan *et al*, 2016). During zebrafish neural rod development, cell adhesion and cell division align to allow an organised structure to arise from dynamically reorganising cells (Symonds & Buckley *et al*, 2020) and loss of N-cadherin results in mis-oriented cell divisions and a disrupted

apical domain (Žigman *et al*, 2011). Once apical proteins fuse with the apical membrane, proteins associated with junctions such as Cadherin, PAR-3 and ZO-1 are then cleared from the apical surface and instead form the apical-lateral junctions, as demonstrated in several different epithelial systems (Morais-de-Sa *et al*, 2010; Symonds & Buckley *et al*, 2020; Kim *et al*, 2021).

Whilst we have demonstrated that E-cadherin directs AMIS localisation, we do not yet have a full explanation for why AMIS proteins localise at the central-most point of cell–cell contact in the absence of divisions, despite E-cadherin localisation all along the cell–cell interface. As mentioned, PAR-3 has been shown to directly bind to the transmembrane JAMs and Nectin proteins (Ebnet *et al*, 2001; Itoh *et al*, 2001; Takekuni *et al*, 2003). However, these proteins localised to the AMIS later than PAR-3 and were not necessary for AMIS localisation (Fig 4). PAR-3 and PAR-6 have been found to be directly recruited to Cadherin proteins within endothelial cells (Iden *et al*, 2006). Recently, opposing actin flows in migrating cells as they first encounter each other were found to be responsible for regulating the first AJ deposition via tension-mediated unfolding of a-catenin and further clustering of surface E-cadherin molecules (Noordstra *et al*, 2021). Together, this could provide an explanation for how the first contacts between cells could act as an apical "seed," therefore defining the position of the AMIS within multi-cellular tissues. This could, therefore, explain why we have previously seen an upregulation of N-cadherin at the zebrafish neural rod midline, where cells growing from either side of the organ primordium meet (Symonds & Buckley *et al*, 2020). However, it is still unclear how this might regulate the subcellular localisation of the AMIS to the centre of cell–cell contacts. Our FRAP results demonstrate that E-cadherin is more stable at the central-most point of contact between two adhering cells (Fig 3J and K). This might suggest that E-cadherin is more stably bound via its downstream partners to the internal actin cytoskeleton at this point, which might help to stabilise or to localise AMIS proteins. In line with this hypothesis, previous publications have demonstrated an upregulation of phosphorylated MYOSIN-II (p-MLC) at the AMIS (Molè *et al*, 2021), which is suggestive of higher actomyosin-mediated tension. Uncovering the mechanisms directing adhesion-dependent AMIS localisation precisely to the midpoint of cell–cell adhesions will be an interesting area for future studies. In addition, a recent study of chick neural tube polarisation (where N-cadherin is the dominant Cadherin) has demonstrated that the interaction of β-catenin with pro-N-cadherin in the Golgi apparatus is necessary for the maturation of N-cadherin, which is in turn important for apical-basal polarity establishment (Herrera *et al*, 2021). This provides the possibility that the polarised Golgi apparatus that we observe in the mESC clusters might be directionally delivering mature E-cadherin to the central-most region of cell–cell contact.

## In the absence of an AMIS, lumens form via "closure" rather than hollowing

The centralised localisation of an AMIS appears necessary to enable lumen hollowing within multi-cellular clusters. *Cdh1* KO cells lack AMIS localisation at the 24-h AMIS stage (Fig 3). However, they still retain their apico-basal polarity axis (as denoted by Golgi apparatus and centrosome localisation, Fig 3G–I) and form apico-lateral

junctions at luminal stages of development (Fig 6). Therefore, *Cdh1* KO cells appear to still make an apical membrane (presumably directed by ECM-mediated signalling) but do so more slowly than in WT cells and without going through a centralised AMIS stage. This suggests that the role of E-cadherin in *de novo* polarisation is specifically to localise the AMIS, which enables the integration of individual cell apical domains to a centralised region preceding lumen hollowing. The lack of a centralised AMIS in E-cadherin deficient cells could also explain the multiple-lumen (but otherwise polarised) phenotypes previously seen in E-cadherin deficient MDCK cells cultured on collagen (Jia *et al*, 2011). Although the other adhesion molecules we have tested (P-cadherin, JAM-A and Nectin-2) did not contribute to centralised AMIS formation, mESCs cultured in Matrigel and mouse inner cell mass cells only become fully epithelialised and start to generate the central cavity once they have exited pluripotency and there are multiple cells in the structures (Shahbazi *et al*, 2017; Kim *et al*, 2021). Thus, whilst E-cadherin appears to be essential for AMIS localisation, other adhesion molecules may be important at later polarisation and lumenogenesis stages.

A surprising observation was the ability of *Cdh1* KO mESC clusters, in the absence of AMIS localisation, to instead form "lumen-like" structures via a "closure" process (summarised in Fig 7B). Our movies of *Cdh1* KO cell clusters (Movies EV3 and EV4) confirmed conclusions from fixed data (Fig 6) that *Cdh1* KO cell clusters first generate a polarised, open cup-shape cavity, before "closing." Due to phototoxicity, we only had limited sample size and movie lengths; thus, we were not able to fully exclude the possibility that the hollowing lumenogenesis occurs to some small extent in parallel, but our data are not suggestive of hollowing lumenogenesis in the *Cdh1* KO cell clusters. We do not currently know the mechanism by which such "closure" occurs in *Cdh1* KO cell clusters. However, the presence of F-actin and p-MLC-rich cable-like structures in "cup"-shaped open cavities is potentially suggestive of a contractile process (Fig EV5). Understanding the relative roles of mechanics in localisation of the AMIS and in "opening" vs. "closing" tubes is an important future research goal, as is the potential role of cell geometry in mediating such differences. Additionally, collective cell migration could play a role in this "closure" mechanism. Collective inwards migration of cells caused lumen formation via a folding mechanism when MDCK monolayers were overlaid with a soft collagen gel (Ishida *et al*, 2014). A similar collective process could be occurring in the *Cdh1* KO cell clusters from our study, which were cultured in a soft (10%) Matrigel and formed loosely connected cell clusters, which then "closed" to make a centralised lumen.

In summary, our work suggests that Cadherin-mediated cell–cell adhesion directs AMIS localisation during *de novo* polarisation of epithelial tubes and cavities. Our work also suggests that ECM is insufficient to direct AMIS localisation in the absence of Cadherin. In parallel with the well-described role of the midbody in tethering apical proteins, this suggests that there are two, interrelated mechanisms of AMIS localisation: cell adhesion and cell division. The alignment of these cellular processes allows for redundancy in the system and provides an explanation for how an organised epithelial structure can arise within the centre of a proliferating organ primordium.

# Material and Methods

### Cell cultures and treatment

Cell lines used in this study are listed in Table 1. mESC carriers were maintained in Feeder Cell medium in Corning cell culture dishes precoated with 0.1% Gelatin (ES-006-B, Sigma-Aldrich), at 37°C suppled with 5% $CO_2$ at one atmospheric pressure. The culture medium was renewed every 3 days. The cells were trypsinised when reaching confluency to be passaged or subjected to experiments. The cells were regularly checked to be mycoplasma-contamination-free.

For 3D cultures of wild-type and *Cdh1* KO mESCs, 20 µL of Matrigel (356231, Corning, Lot 354230, 354234, 356231) was spread evenly to the bottom of each well in a µ-slide 8-well dish (80821, Ibidi). The dish was left on ice for 10 min to flatten the Matrigel surface and then was left at 37°C for 10 min to solidify the Matrigel. mESCs were trypsinised, pipetted thoroughly and passed through a cell strainer (431750, Corning) to isolate cells into single cells. Singled mESCs were suspended in the N2B27 medium and seeded onto the solidified Matrigel. The seeded density was: control, 14 cells/mm$^2$; mitomycin C-treated, 227 cells/mm$^2$. The cells were left at 37°C for 15 min when over 95% of the cells attached to the Matrigel, then the culture medium was renewed to 10% Matrigel/N2B27 medium with or without 2i/LIF.

For control and *Cdh1* KO mESC chimeric cluster cultures, wild-type and *Cdh1* KO mESCs were mixed at 1:4 ratio and co-cultured in 2D in the Feeder Cell Medium. They were then treated with mitomycin C for 2 h, trypsinised and seeded for 3D Matrigel culture at 227 cells/mm$^2$.

For mESC cultured in agarose, 5,000 control or 125,000 mitomycin C-treated cells were suspended in a 37°C warmed 20 µL 0.5% low melting point agarose (16520, Invitrogen) droplet at the bottom of the µ-slide 8-well dish. The dish was left at room temperature for 5 min to solidify and topped with the N2B27 medium. The cells were then cultured at 37°C, 5% $CO_2$ until analysis.

For cells cultured on E-cadherin and fibronectin-coated glass, the µ-slide 8-well dish was incubated with nitrocellulose/methanal at 37°C for 3 h and left to air dry. The dish was then incubated with 40 µg/ml mouse E-cadherin recombinant protein (8875-EC-050, Bio-Techne) or 40 µg/ml fibronectin (F1141, Sigma-Aldrich) at 4°C overnight. The dish was briefly washed with water. Mitomycin C pre-treated ES-E14 cells were seeded onto the dish at 14 cells/mm$^2$ in N2B27 medium. The cells were allowed to attach to the glass at 37 °C for 1 h, and then the medium was renewed to N2B27 medium with 20% Matrigel. The cells were fixed 24-h post-Matrigel introduction.

For mitomycin C treatment, the cells were incubated with 10 µg/ml mitomycin C (J63193, Alfa Aesar) in culture media at 37°C for 2 h. The mitomycin C-contained media were removed, and the cells were washed with PBS briefly. Then, the mitomycin C-treated cells were trypsinised and subjected to further experiments.

Mouse PAR-6B coding DNA sequence (cDNA, GenBank: BC025147.1) was assembled with mCherry by Gibson assembly, and LifeAct-Ruby cDNA were sub-cloned from an existing pRN3P-LifeAct-Ruby plasmid. The cDNAs were cloned into pDONR221 plasmid and introduced into the PB-Hyg-Dest plasmid using Gateway technology (Thermo Fisher Scientific). The PB-Hyg-Dest-mCherry-PAR-6B or the PB-Hyg-Dest-LifeAct-Ruby plasmid was co-transfected with the piggyBac plasmid using Lipofectamine 3,000 to generate Hygromycin B resistant stable cell lines. The mCherry-PAR-6B or LifeAct-Ruby expressing mESC stable cell line was created via 10 µg/ml Hygromycin B selection and single-cell colonies expansion. Primers used for cloning are listed in Table 2.

### Compositions of cell culture media

Feeder Cell Medium: DMEM (41966, Thermo Fisher Scientific), 15% FBS (ES-009-B, Sigma-Aldrich), penicillin–streptomycin (15140122,

**Table 1. List of cell lines used in this study.**

| Name | Description | Animal strains | Source | RRID | References |
|---|---|---|---|---|---|
| ES-E14 | Wild-type mouse embryonic stem cells | 129P2/Ola mice | Cambridge Stem Cell Institute | CVCL_C320 | Hooper *et al* (1987) |
| W4 | Wild-type mouse embryonic stem cells | 129S6/SvEvTacArc mice | Gifted from Shukry Habib at King's College London | CVCL_Y634 | Auerbach *et al* (2000) |
| *Cdh1* KO | E-cadherin knockout mouse embryonic stem cells | 129S6/SvEvTacArc mice | Gifted from Lionel Larue at Institute Curie | | Larue *et al* (1996) |
| mCherry-PAR-6B ES-E14 | ES-E14 mouse embryonic stem cells expressing mCherry-PAR-6B | 129P2/Ola mice | Generated in this study | | |
| E-cadherin-eGFP ES-E14 | ES-E14 mouse embryonic stem cells expressing E-cadherin-eGFP | 129P2/Ola mice | Published in Molè *et al* (2021) | | Molè *et al* (2021) |
| LifeAct-mRuby ES-E14 | ES-E14 mouse embryonic stem cells expressing LifeAct-mRuby | 129P2/Ola mice | Generated in this study | | |
| LifeAct-mRuby *Cdh1* KO | E-cadherin knockout mouse embryonic stem cells expressing LifeAct-mRuby | 129S6/SvEvTacArc mice | Generated in this study | | |

**Table 2.   Primers used for cloning.**

| **Forward: attB1-N-mCherry** |
| --- |
| 5-GGGGACAAGTTTGTACAAAAAAGCAGGCTTCGCCACCATGGTGAGCAAGGG-3 |
| **Reverse: C-mCherry-N-PAR6B** |
| 5-TGCCGGTGGCCGCGGTTCATCGGATCCCCCGGGCTGCAGGA-3 |
| **Forward: C-mCherry-N-PAR6B** |
| 5-TCCTGCAGCCCGGGGGATCCGATGAACCGCGGCCACCGGCA-3 |
| **Reverse: C-PAR6B-attB2** |
| 5-GGGGACCACTTTGTACAAGAAAGCTGGGTCTAGCAGATGATGTCGTCCTCGT-3 |
| **Forward: attB1-N-LifeAct** |
| 5-GGGGACAAGTTTGTACAAAAAAGCAGGCTTCGAAGGAGATAGAACCATGGGCGTGGCCGACCTGAT-3 |
| **Reverse: C-mRuby-attB2** |
| 5-GGGGACCACTTTGTACAAGAAAGCTGGGTGTTACCCTCCGCCCAGGCCGG-3 |

Thermo Fisher Scientific), GlutaMAX (35050061, Thermo Fisher Scientific), MEM non-essential amino acids (11140035, Thermo Fisher Scientific), 1 mM sodium pyruvate (11360070, Thermo Fisher Scientific) and 100 μM β-mercaptoethanol (31350-010, Thermo Fisher Scientific). To maintain cells in pluripotency, 2i/LIF (1 mM MEK inhibitor PD0325901, 13034, Cayman Chemical; 3 mM GSK3 inhibitor CHIR99021, 13122, Cayman Chemical; and 10 ng/ml leukaemia inhibitory factor, LIF, A35933, Gibco) was added to the Feeder Cell medium to preserve naïve pluripotency. N2B27 Medium: 1:1 mix of DMEM F12 (21331-020, Thermo Fisher Scientific) and neurobasal A (10888-022, Thermo Fisher Scientific) supplemented with 2% v/v B27 (10889-038, Thermo Fisher Scientific), 0.2% v/v N2 (17502048, Gibco), 100 μM β-mercaptoethanol (31350-010, Thermo Fisher Scientific), penicillin–streptomycin (15140122, Thermo Fisher Scientific) and GlutaMAX (35050061, Thermo Fisher Scientific).

### siRNA transfection

To achieve protein knockdowns, siRNA was transfected using Lipofectamine RNAiMAX according to the manufacturer's instructions. mESCs cultured in 2D on the gelatin were transfected with 100 nM pre-designed siRNA (s63752, Silencer Select). siRNA target mRNA sequences were as follows: E-cadherin, GAAGAUCACGUAUCGGAUU; P-cadherin, CGAAAGAGAGAGUGGGUGA; JAM-A, GCCUUUGAUAGUGGUGAAU; Nectin-2, GGACUACUGAAUUCUUUUA. Two days after the first transfection (Fig EV4A), the cells were cultured with or without mitomycin C and seeded into 10% Matrigel. The cells were cultured in Matrigel with Lipofectamine RNAiMAX and 100 nM siRNA for another 24 h and then were subjected to further experiments and analysis.

### Immunofluorescence

Cells cultured in a μ-slide 8-well dish were fixed with 4% paraformaldehyde (J61899, Alfa Aesar) for 30 min at room temperature and then were permeabilised with 0.5% Triton X-100 for 15 min at room temperature. The cells were blocked with the incubation buffer (0.5% BSA, 0.1% Tween in PBS) for 2 h and then were incubated with primary antibodies diluted in the

incubation buffer at 4°C overnight on an orbital shaker. The primary antibodies were washed off with PBS and then were incubated with secondary antibodies diluted in the incubation buffer at room temperature for 2 h. The secondary antibodies were washed off with PBS. Samples cultured in Matrigel were kept in PBS; samples cultured in agarose were sealed in 200 μl 0.5% agarose. The samples were imaged shortly after. Antibodies and dilutions were as listed in Table 3.

### Microscope imaging

Live cell imaging was carried out on the PerkinElmer UltraVIEW spinning disc system fitted on an Olympus IX80 confocal microscope with a 37°C and 5% $CO_2$ chamber. Images were captured with the 40× 1.3 NA (oil) UPLAN objective, 2× Hamamatsu Orca-R2 CCD camera and Volocity 3.7.1 software. The cells were imaged at 2 μm z-step size and 30-min time intervals. Fixed samples were imaged on the Leica SP8 confocal microscope with the 40× 1.3 NA (oil) or 63× 1.4 NA (oil) Plan Apo objective and LAS X 3.7.4 software. The cells were imaged at 0.3 μm z-step size and 2× line average. FRAP was performed on the Zeiss LSM-900 confocal microscope with 63× 1.40 NA (oil) Plan Apo objective and ZEN Blue 2.1 software equipped with a 37°C heated stage.

### Image and data analysis

The central section images were projected from raw images in the Fiji software by maximum-value projection of the whole z stacks to produce the 3D projection images, or of the central three images of the z stacks to produce the central section images. The whole z stacks projections were applied to count cell cluster percentages. The central section images were applied for line-scans or analysis of region of interest to determine protein signals.

The mESCs with positive or negative protein centres were manually determined and counted. The percentage was calculated with the number relative to the number of total cell clusters captured in each condition. The mean percentages from three independent experiments were compared with Student's *t*-test or one-way ANOVA specified in figure legends using the GraphPad Prism software. Sample sizes are specified in figure legends.

**Table 3. List of primary and secondary antibodies used for immunofluorescence in this study.**

| Primary antibodies | | | | | |
|---|---|---|---|---|---|
| **Protein** | **Catalogue number** | **Supplier** | **Type** | **Specie** | **Concentrations** |
| E-cadherin | ECCD-2 | Invitrogen | Monoclonal | Rat | 2 µg/ml |
| GM130 | 610822 | BD Biosciences | Monoclonal | Mouse | 1:300 |
| Nanog | ab80892 | Abcam | Polyclonal | Mouse | 1:300 |
| JAM-A | sc-52688 | Santa Cruz | Monoclonal | Rat | 1:300 |
| Nectin-2 | 502-57 | HycultBiotech | Monoclonal | Rat | 1:200 |
| OTX2 | AF1979 | R&D Systems | Polyclonal | Goat | 1:300 |
| PAR-3 | 07-330 | Merck Millipore | Polyclonal | Rabbit | 1:100 |
| P-cadherin | AF761 | R&D Systems | Polyclonal | Goat | 1:500 |
| Podocalyxin | MAB1556 | R&D Systems | Monoclonal | Rat | 3.3 µg/ml |
| Γ-Tubulin | T6557 | Sigma-Aldrich | Monoclonal | Mouse | 1:250 |
| ZO-1 | 61-7300 | Invitrogen | Polyclonal | Rabbit | 1:500 |

| Secondary antibodies and reagents | | | |
|---|---|---|---|
| **Name** | **Supplier** | **Target/catalogue number** | **Concentrations** |
| Alexa Fluor 488 | Invitrogen | Rat | 1:500 |
| Alexa Fluor 488 | Invitrogen | Rabbit | 1:500 |
| Alexa Fluor 488 | Invitrogen | Goat | 1:500 |
| Alexa Fluor 546 | Invitrogen | Mouse | 1:500 |
| Alexa Fluor 555 | Invitrogen | Goat | 1:500 |
| DyLight 550 | Invitrogen | Rat | 1:500 |
| CF 633 | Insight Biotech | Phalloidin | 6.6 pM |
| 4′,6-diamidino-2-phenylindole, DAPI | Sigma | D8417 | 0.5 µg/ml |

To quantify PAR-3 signal along the cell–cell interface, a 0.8 µm width line was drawn along the cell–cell interface by using F-actin (labelled by Phalloidin) or E-cadherin signal as the path. PAR-3 and F-actin pixel values along the path were extracted. The two peaks of F-actin signals at two ends of the path were determined as the start and end of the cell–cell interface, and the positions in-between were defined as 1× cell–cell interface. The corresponding PAR-3 pixel to the F-actin peak positions was identified, and the PAR-3 line profile between the two positions was sectioned into 20 sections. PAR-3 pixel value in each section was averaged to be the PAR-3 signal in 5% of the cell–cell interface. The values from 10 to 20 cells were plotted as line graphs.

To compare the level of PAR-3 at the centre and outer regions in multi-cell mESC clusters, a 4-µm-diameter circular area was created in the Fiji software and placed over the multi-cellular joint in a cell cluster, and the average PAR-3 fluorescence values in the circle were measured. The boundary of the cell cluster was drawn by the free-hand tool in Fiji by using the correspondent F-actin signal, and the average PAR-3 fluorescence values between the circle and the boundary were measured. The ratio between the PAR-3 values in the circle and outer regions was then calculated.

To generate the PAR-3 signal heatmap, based on the F-actin signal, a squared region of interest was created in the Fiji software for every cell, which the main cell body of a cell was fitted into. The PAR-3 image in each squared region was transformed into a 10 × 10-unit matrix by using the R-language. Each unit was created by averaging the PAR-3 fluorescence intensity in every 10% length along the *X* or *Y* axis of the squared region of interest extracted from the original image. A serial of images from one condition of an experiment was stacked, and the averaged pixel value at each position was calculated to generate an averaged 10 × 10 matrix. The final matrix was transformed into a heatmap in the GraphPad Prism software.

To compare the level of PAR-3 in the core areas in the cells culture on the glass, a 6-µm-diameter circular area of interest was created in the Fiji software to cover the PAR-3 core area in a cell, and the average PAR-3 fluorescence values in the circle were measured. The boundary of the cell was drawn by the free-hand tool in Fiji, and the average PAR-3 fluorescence values in the boundary were measured. The ratio between the PAR-3 values in the circle and inside the boundary was then calculated.

Sample sizes and statistical analysis are detailed in the figure legends. No blinding was done for the experiments in this study. Instead, we included direct continuous measurements (e.g. of fluorescent intensity) alongside categorical analyses. To assess AMIS localisation independently to cell division, we did not include control cells that were undergoing mitosis (identified by chromosome and cell morphologies) into the quantifications. We checked that excluding this data made no difference to the outcome of the analyses.

### FRAP

FRAP of E-cadherin-eGFP was performed to measure E-cadherin dynamics at the cell–cell interface. An E-cadherin-eGFP expressing stable ES-E14 line was used to perform the experiment (Molè et al, 2021). After 24-h culturing in Matrigel, two-cell clusters with long axis parallel to the surface of the culture dish, hence the cell–cell interfaces were aligned to the axis of the objective were used for FRAP. A region of interest (ROI) of $2 \times 1$ μm along $xy$-axis was chosen at the approximately centre-most region or side regions at each cell–cell interface. The ROI in a series of $z$-axis stacks with 0.3-μm intervals totalled to 2.1 μm in depth was bleached with three iterations of the 488 nm laser with 100% transmission. This resulted in a photobleaching of over 80% in a $2 \times 2.1$ μm ($xz$-axis) region at the cell–cell interface. Time-lapse images were acquired before (three frames) and after (30 frames) photobleaching with an interval of 10s per frame. Average fluorescence intensity values $F(t)$ in the bleached area within the centre z-stack were analysed with ImageJ. The mean values of the three frames before bleaching were used as the pre-bleached value $F(i)$. The value of the first frame after bleaching was defined as $F(0)$. FRAP values were then calculated and plotted over time in Fig 3J as:

$$\text{FRAP} = \frac{F(t) - F(0)}{F(i) - F(0)}$$

The FRAP values were fitted using a non-linear regression and the exponential one-phase association model using Y0 = 0 and where mobile fraction corresponds to the plateau value in the GraphPad Prism software. For Fig 3K, the mobile fraction from each FRAP profile was pooled and compared between conditions.

## Data availability

Image data are accessible in the BioImage Archive, accession number S-BIAD473 (https://www.ebi.ac.uk/biostudies/bioimages/studies/S-BIAD473?query=S-BIAD473). The R-language code for generating the PAR-3 heatmap in Fig 5D is accessible at: https://github.com/Buckley-Lab-opto/Liang-2022.

**Expanded View** for this article is available online.

## Acknowledgements

We are grateful to Jon Clarke and Ben Steventon for critical reading of the manuscript and other members of the Buckley and Zernicka-Goetz laboratories for scientific discussion, especially Matteo Molè and Marta Shahbazi. Thank you to the Shukry Habib laboratory for kindly gifting the W4 cells and the Lionel Larue laboratory for kindly gifting the Cdh1 KO cells. Thank you to the Cambridge Advanced Imaging Centre and the Ewa Paluch laboratory for help and access to confocal microscopy. This research was financially supported by: CEB—the Wellcome Trust and Royal Society (Sir Henry Dale Fellowship grant no. 208758/Z/17/Z and Dorothy Hodgkin Fellowship grant no. DH160086), XL—European Union's Horizon 2020 programme (Marie Skłodowska-Curie Individual Fellowship grant no. 844330), the Issac Newton Trust and Leverhulme Trust (Leverhulme Early Career Fellowship grant no. ECF-2019-175). MZG - The Wellcome Trust (207415/Z/17/Z) and ERC (669198). For the purpose of open access, the authors have applied a

## Author contributions

**Xuan Liang:** Conceptualization; data curation; formal analysis; funding acquisition; investigation; visualization; methodology; writing – original draft; writing – review and editing. **Antonia Weberling:** Investigation; methodology. **Chun Yuan Hii:** Data curation; formal analysis. **Magdalena Zernicka-Goetz:** Resources; supervision; funding acquisition. **Clare E Buckley:** Conceptualization; resources; data curation; supervision; funding acquisition; methodology; writing – original draft; project administration; writing – review and editing.

## Disclosure and competing interests statement

The authors declare that they have no conflict of interest.

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
