## [Review Process File · The EMBO Journal]

E-cadherin mediates apical membrane initiation site localisation during de novo polarisation of epithelial cavities

Xuan Liang, Antonia Weberling, Chun Yuan Hii, Magdalena Zernicka-Goetz, and Clare Buckley
DOI: 10.15252/emboj.2022111021

Corresponding author(s): Clare Buckley (ceb85@cam.ac.uk)

Review Timeline:

Transfer from Review Commons:	25th Feb 22
Editorial Decision:	9th Mar 22
Revision Received:	7th Jun 22
Editorial Decision:	8th Jul 22
Revision Received:	18th Jul 22
Accepted:	21st Jul 22

Editor: William Teale

Review
COMMONS

Transaction Report: This manuscript was transferred to The EMBO JOURNAL following peer review at Review Commons.

Review #1

1. Evidence, reproducibility and clarity:

Evidence, reproducibility and clarity (Required)

Summary:

Formation of tubes in a developing organism may arise from the closure of a pre-existing polarized epithelium or from de novo polarization and cavity formation in group of dividing cells. The concept of apical membrane initiation site (AMIS) refers to the fact that polarity proteins as PAR3 accumulate at a point where the apical membrane will be created. This accumulation occurs as early as the two cell stage. Previous reports have demonstrated the importance of the division process in defining this AMIS, however, in the present work the authors in vitro 3D cultures of mESC to report a mitosis independent mechanism that creates an AMIS, induces the polarization of groups of two or more cells, and permits the formation of a central cavity. The report shows that the mechanism is fully dependent on the polarized accumulation of E-cadherin at the cell membrane in contact with the other cells. Moreover, the mechanism does not require mitosis or interaction with the extracellular matrix.

Major comments:

The main objective of the work is to demonstrate that AMIS creation and cavity formation can be mitosis independent and that it is dependent on the accumulation of E-cadherin at the midline between two cells in contact. To demonstrate these objectives, the authors perform 3D cultures of mESC. To rule out the requirement of mitosis the authors perform cultures that are treated with mitomycin C and the purify single cells that are cultured again. The authors show time-laps experiments demonstrating that individual cells that do not divided create an AMIS when they contact one to each other. With this cultures they demonstrate that the process does not require an interaction with ECM (provided by the matrigel) but requires E-cadherin, to demonstrate, that they use E-cadherin KO cells (the same line where E-cadherin has been deleted). The work is well written and the objectives very clear. The technology used and the experiments done are adequate and sufficient to accomplish the proposed objectives and the results obtained clearly support the conclusions reached. The methods are well explained and transparent to be reproduced elsewhere and the number of replicas and the statistical methods applied seem corrects to me, although I am just a biologist, not a mathematician. Although the objectives of the work, that are: to demonstrate that AMIS formation can be independent of mitosis and that AMIS requires E-cadherin, there are parts of the results that could be farther studied or at least discussed more thoroughly. Firstly, the authors show that in non-dividing cells an AMIS is formed at the first contact site between the two cells, they also show that in the absence of E-cadherin the cell maintains the polarization of centrioles and Golgi apparatus, in spite that no AMIS is formed, this indicates that the deposition of E-cadherin at the midline membrane is part of a more global polarization event that most likely is initiated by the a directional activity of the Golgi apparatus that may direct the delivery of mature E-cadherin in that particular direction, initiating or maintaining the basis for an AMIS, since recent work (already cited in the manuscript) has demonstrated the importance of cadherin maturation for polarity establishment and maintenance (Herrera et al, 2021), the actual results should be farther discussed in this context. Secondly, it was previously shown that in different epithelia, upon cell-cell contact, the aPKC complex (that includes Par3 and Par6) is recruited early to the contact site where with the participation of Cdc42, aPKC is activated generating an initial spot-like adherent junction (AJs) (Suzuki et al., 2002). In that case it is thought to be mediated by a direct interaction between the first PDZ domain of PAR-3 and the C-terminal PDZdomain-binding sequences of immunoglobulin-like cell adhesion molecules: JAM-1 and nectin-1/3 (Fig. 3) (Ebnet et al., 2001; Itoh et al., 2001; Takekuni et al., 2003). Thus it would be interesting to know if AMIS formation in absence of cell division depends on JAM-1 or nectin and whether JAM-/Nectin signalling is sufficient to initiate the Golgi and centriole polarization and which is the mechanism governing it.

****Minor comments:****

As I mentioned before, the paper is well presented and very clear, yes it is simple, but simple is always better, no complicated graphics or letterings, thank you.

Although in my opinion the work is very well written, I have to admit that I am not qualified to evaluate the literary style of the work since English is not my mother tongue, also I have not reviewed typographical errors since I think that is the work of the editorial, not of scientific reviewers.

Please include the full reference of all the antibodies used, including the company and not just the catalog number

****Quoted references:****

Ebnet, K., Suzuki, A., Horikoshi, Y., Hirose, T., Meyer Zu Brickwedde, M. K., Ohno, S. and Vestweber, D. (2001). The cell polarity protein ASIP/PAR-3 directly associates with junctional adhesion molecule (JAM). *EMBO J.* 20, 3738-3748.

Itoh, M., Sasaki, H., Furuse, M., Ozaki, H., Kita, T. and Tsukita, S. (2001). Junctional adhesion molecule (JAM) binds to PAR-3: a possible mechanism for the recruitment of PAR-3 to tight junctions. *J. Cell Biol.* 154, 491-497.

Takekuni, K., Ikeda, W., Fujito, T., Morimoto, K., Takeuchi, M., Monden, M. and Takai, Y. (2003). Direct binding of cell polarity protein PAR-3 to cell-cell adhesion molecule nectin at neuroepithelial cells of developing mouse. *J. Biol. Chem.* 278, 5497-5500

Suzuki, A., Ishiyama, C., Hashiba, K., Shimizu, M., Ebnet, K. and Ohno, S. (2002). aPKC kinase activity is required for the asymmetric differentiation of the premature junctional complex during epithelial cell polarization. *J. Cell Sci.* 115, 3565-3573.

2. Significance:

Significance (Required)

The paper describes for the first time that contrary to what was previously believed an AMIS can be generated without a cell division. This is very important because it opens the possibility that the mechanisms that originate the biologic cavities are in fact not really how we believed.

The work is of interest of all cell biology scientists, specially working in developmental biology, cancer research.

My particular field of expertise is cell biology and signaling, always applied to particular events as nervous system development or cancer, in particular I am interested in Wnt/b-catenin and Sonic Hedgehog pathways.

3. How much time do you estimate the authors will need to complete the suggested revisions:

Estimated time to Complete Revisions (Required)

(Decision Recommendation)

Less than 1 month

Review #2

1. Evidence, reproducibility and clarity:

Evidence, reproducibility and clarity (Required)

****Summary:****

The importance of cell division and the post-mitotic midbody in the establishment of the apical membrane initiation site (AMIS) is quite well established. However, there are observations hinting to a cell division-independent mechanism of the AMIS formation. The authors hypothesized that cell adhesion involving E-cadherin could direct the site for AMIS localisation during de novo polarisation.

As model system the authors used mouse embryo stem cell (mESC) culture in Matrigel, which has been used as an in vitro model for the de novo polarisation of the mouse epiblast. The slow lumen formation in culture allows for a relatively clear separation of the stages of de novo polarisation. This enables to study the initiation of apico-basal polarity of embryonic cells alongside the first cell-cell contacts between isolated cells and small cell clusters. Here, the goal was to determine the role of cell adhesion, and in particular E-cadherin, in mESC AMIS localisation.

****Major comments:****

1. Mitomycin C is commonly used to block cell division, however, what it does is it blocks DNA replication, and the blockage of cell division is a consequence. It could have other effects than only blocking cell division. What about using a mitosis blocker? It would be good to have a second way in addition to mitomycin C treatment of confirming that the results support the conclusion that cell division is dispensable for AMIS localization, as the full work builds up on that first observation and this experimental setup is carried on through the manuscript.

2. Many clusters at higher cell stage that are shown (e.g. Fig 2C, 3A), look like they are in the perfect 4-cell stage after cell division. Movie 1 does not show that, only 2 cells are clustering. Movie 2 shows how two 2-cell clusters form a 4-cell cluster, however, that does not look as "perfect" as the 4-cell stages shown in the figures, which look as said more like 4-cell stages resulting from cell division. Maybe the authors could provide more movies that show the 2-cell cluster doublets (4-cell clusters)? Also, the puncta relocalization to the cell-cell contacts in the 2-cell cluster doublet is not so very clear in the movie 2. Maybe the authors have more movies for 2-cell cluster doublets?

3. On page 4 the authors observe: "Whilst homogenous control doublets localised PAR-3 to the central region of the cell-cell interface, heterogeneous chimeric doublets did not localise PAR-3 centrally (Figure 3D,E). Golgi and centrosome localisation towards the cell-cell interface suggested that the overall axis of polarity was maintained, even in the absence of both cell division and E-CADHERIN (Figure 3F-H & S2C)." This means that the axis of polarity is established without E-cadherin and without the AMIS. So, the cell-cell contact site itself is able to establish polarity, but not localize the AMIS.

In line with this, the authors state: "The results also demonstrate that ECM in the absence of E-CADHERIN is insufficient for AMIS localisation." So, without E-cadherin, there is no AMIS, the ECM cannot establish it, but polarity is still established. On p. 6, it is then concluded that E-cadherin and centralised AMIS localisation are not required for apical membrane formation, but that they promote its formation earlier and more efficiently in development. In the discussion, however, the authors then state in a rather contrary manner "Our results therefore suggest that CADHERIN-mediated cell-cell adhesion may provide the symmetry-breaking step required for AMIS localisation during de novo polarisation." However, cells are polarized and have an apical membrane (Figure 3F-H & S2C) without cadherin and the AMIS, so how can this be the symmetry-breaking step for de novo polarization? Later on, in line with the earlier statement that E-cadherin and the AMIS location are not required for apical membrane formation, the authors then refine the previous statement: "the role of E-CADHERIN in de novo polarisation is specifically to localise the AMIS, which enables the integration of individual cell apical domains to a centralised region preceding lumen hollowing." This seems to be more likely to me than the CADHERIN-mediated cell-cell adhesion as the symmetry-breaking step. I

therefore disagree in this point with their overall summary on p. 8, and find this a bit confusing.

4. As Wild-type mESCs (ES-E14) were purchased from Cambridge Stem Cell Institute and Cdh1 KO mESCs were gifted from a lab, it would be good to (genetically) characterize the cell lines because, apparently, they do not have the same origin and the KO cells were not derived from the parental mESCs. Alternatively, a control experiment with knockdown of Cdh1 in the purchased mESCs could be done, even if that would not lead to a complete knockdown, to make sure that the observed effects are the same as with the Cdh1 KO cell line.

5. The existing data is carefully analyzed with appropriate statistics. Replicates are sufficient. The conclusions are not yet fully justified, as discussed.

****Minor comments:****

- Fig S1D: It is labeled "Pard3". While also correct, it should be consistent, i.e. PAR3.

- P. 5 remove (slightly)

- P. 2, p. 7 "Pard3", replace with PAR3

- The antibody lists already contain catalog numbers in case of the primary antibodies, but no suppliers. They should be added, and also specified for the secondary antibodies for better reproducibility.

In general, the text and figures are well prepared and of high quality. The citations appear appropriate.

2. Significance:

Significance (Required)

The work describes the conceptual novelty of cell adhesion as alternative mechanism to cell division for AMIS localization, and in particular E-Cadherin as being required for AMIS positioning. It is still unclear why the AMIS is centered and the localization of cadherin is equal along cell-cell contacts (Fig 2C, S1E). How do Cadherin localization dynamics look like during the clustering of two cells? During cell division in a MDCK cyst (which is where my expertise lies), cell adhesion has to be partially removed during cytokinesis and abscission, and then be installed again, basically like a new cell-cell contact. Thus, could E-cadherin focus ("trap") the "AMIS initiation seed", rather than direct binding of PAR3 /PAR6 to cadherin as discussed by the authors, since E-cadherin is localized along the whole contact site and not centred? Could the unknown "apical seed" (which in cell division is the midbody) be trapped by cell adhesion? Could this be a common mechanism between cell division- and cell adhesion-driven AMIS localization? This finding could therefore have an even broader impact. What are the author's thoughts? While my speculation might be wrong, it might be worth hypothesizing on the connection between the role of E-cadherin in the two ways of AMIS localization.

Another novelty is the observation that polarity and cavities form later on in development independently of E-cadherin and an AMIS. This type of mechanism should be discussed further and put more into perspective with the literature.

The work describes a new mechanism which could be of broad importance in developmental biology. I therefore think that this work is highly significant.

3. How much time do you estimate the authors will need to complete the suggested revisions:

Estimated time to Complete Revisions (Required)

(Decision Recommendation)

Between 1 and 3 months

Review #3

1. Evidence, reproducibility and clarity:

Evidence, reproducibility and clarity (Required)

In this manuscript, Xuan Liang and collaborators shed light on how the precise localisation of the apical membrane initiation site (AMIS), necessary for organised lumen formation, is directed at the single-cell level. By characterising de novo polarising mouse embryonic stem cells (mESCs) cultured in 3D, the authors have uncovered a division-independent mechanism of de novo polarisation and AMIS localisation based on adhesion molecules. More precisely, they suggest that E-CADHERIN-mediated cell-cell adhesion may provide the symmetry-breaking step required for AMIS localisation during de novo polarisation since this molecule alone is sufficient and necessary to drive correct AMIS localisation. Interestingly, a high proportion of E-Cadherin knock-out (Cdh1 KO) mESC cell clusters do not hollow but instead generate lumen-like cavities via a closure mechanism. Despite not knowing the mechanism involved in the closure of these lumen-like cavities, the role of E-CADHERIN in de novo polarisation would be associated with initial steps in lumen formation (AMIS formation and localisation) but not in later steps where E-Cadherin knock-out mESC cell clusters can still make an apical membrane but do so more slowly than in WT cells and without going through a centralised AMIS stage.

Altogether, this study supports their previously published zebrafish neuroepithelial cell in vivo analysis, which demonstrated the division-independent localisation of Pard3 and ZO-1 at the neural rod primordial midline (Buckley et al., 2013). The authors have provided a novel mechanism of de novo polarisation and AMIS formation that occurs in vivo and in vitro. For this reason, this is a work with great significance that will undoubtedly be of general interest to the readers of Review commons. Nonetheless, several issues should be addressed before the publication of this manuscript.

1. In figure 4, the authors tried to demonstrate that E-CADHERIN is sufficient for AMIS localisation, independent of ECM signalling and cell division. To this end, they cultured individual division-blocked mESCs onto either E-CADHERIN-FC recombinant protein or FIBRONECTIN pre-coated glass and carried out IHC for PAR-3 after 24 hours in culture. They then performed heatmaps and analysed PAR3 intensity (Fig. 4 D, E). Although the data presented are fascinating and show the effects the authors describe, the authors should improve their sample number and repeat this experimental procedure and analysis at least two more times for their results to be consistent (only 15 cells in one experiment have been used to carry out the statistical analysis described previously).

2. The expression of E-cad is necessary for the proteins that define the apical membrane (Par3, Par6, aPKC) to be located in the AMIS. The results are clear and robust. Even so, I do not think this is sufficient, as the authors claim (headline of page 5, Figure 4). It seems clear that something more than Ecad is needed for the localisation of Par3 in the AMIS because, as the authors indicate in the discussion, Ecad is located along the entire cell-cell junction, while par3 is focused on AMIS. There must be something else that is necessary for the location of Par3. Therefore, the experiment in figure 4 does not prove that E-cad is sufficient but confirms that it is necessary for that location. Another series of experiments would have to be carried out to prove that it is sufficient. This must be clearly stated in the final version of the manuscript.

3. It is clear that E-Cadherin knock-out mESC cell clusters open cup-shaped cavities before generating a lumen-like structure. Fig. 5 presents compelling data about this in vivo lumen formation mechanism without hollowing, though they briefly describe this process. Whilst I am conscious that they do not know the mechanism by which such 'closure' occurs and that this would be suitable for another manuscript, I would strongly suggest including a live movie depicting early stages (before 78:00) of E-Cadherin knock-out cluster development. Many queries arise with this piece of data, as it seems that a small lumen could be forming prior to the cup-shaped cavity.

****Minor points****

1. Fig. 2A: Actin staining could be included to better visualise the spheroids.
2. Fig. 5B is very small, I would recommend them to present it bigger.
3. I would encourage the authors to revise the figures as some have displaced text. (see Fig. 5, 72 hours, Cdh1 KO).

2. Significance:

Significance (Required)

Altogether, this study supports their previously published zebrafish neuroepithelial cell in vivo analysis, which demonstrated the division-independent localisation of Pard3 and ZO-1 at the neural rod primordial midline (Buckley et al., 2013). The authors have provided a novel mechanism of de novo polarisation and AMIS formation that occurs in vivo and in vitro. For this reason, this is a work with great significance that will undoubtedly be of general interest to the readers of Review commons. Nonetheless, several issues should be addressed before the publication of this manuscript. My lab work in lumen formation in 3D organotypic cultures and organoids

3. How much time do you estimate the authors will need to complete the suggested revisions:

Estimated time to Complete Revisions (Required)

(Decision Recommendation)

Between 1 and 3 months

Revision Plan

Manuscript number: RC-2021-01189

Corresponding author(s): Clare Buckley

[The “revision plan” should delineate the revisions that authors intend to carry out in response to the points raised by the referees. It also provides the authors with the opportunity to explain their view of the paper and of the referee reports.]

The document is important for the editors of affiliate journals when they make a first decision on the transferred manuscript. It will also be useful to readers of the reprint and help them to obtain a balanced view of the paper.

*If you wish to submit a full revision, please use our "Full Revision" template. **It is important to use the appropriate template to clearly inform the editors of your intentions.**]*

1. General Statements [optional]

This section is optional. Insert here any general statements you wish to make about the goal of the study or about the reviews.

We would like to thank all the reviewers for their time and for their positive and constructive review of our study. We are happy that they all regard this as a highly significant piece of work. We have addressed their suggestions in the following sections of the Revision plan:

Reviewer 1 Point 1: Section 3

Reviewer 1 Point 2: Section 2

Reviewer 2 Point 1: Section 2

Reviewer 2 Point 2: Section 2

Reviewer 2 Point 3: Section 3

Reviewer 2 Point 4: Section 2

Reviewer 2 Point 5: Section 3

Reviewer 3 Point 1: Section 2

Revision Plan

Reviewer 3 Point 2: Section 3

Reviewer 3 Point 3: Section 4

Reviewers' minor comments: Section 2 and 3

2. Description of the planned revisions

Reply to Reviewer 1 Point 2

The reviewer suggests it would be interesting to know whether there is a role for the proteins JAM-1 or Nectin in AMIS formation and in polarising the Golgi and centrosomes towards the cell-cell contact. Like E-Cadherin, these are transmembrane junctional proteins that are present at the initiation of spot adhesions in epithelial 2D monolayers and are known to be part of a complex network of interactions between PAR-complex, junctional molecules, MAGUK scaffolding proteins and the actin cytoskeleton. Whilst we don't propose to untangle this network here, we agree that it would be interesting to know more about the potential role of JAM-1 and Nectin in initiating polarity in mESC 3D cultures. However, it is important to note that, regardless of whether JAM-1 and Nectin also play a role in polarisation and AMIS formation, our results already demonstrate that E-cadherin-based adhesions are sufficient to **initiate** AMIS localisation. For example, our results from figure 4C-E demonstrate that, in a reductionist system of a single cell plated on E-Cadherin covered glass, a centrally located AMIS still forms. Precisely unravelling the mechanisms by which this happens would be better for a future study (which we have now stated in the Discussion).

Nevertheless, we now have new FRAP data (Figure 3I and J), which demonstrates that E-Cadherin is relatively more stable at the central-most point of contact between two adhering cells. This suggests that E-Cadherin is more stably bound via its downstream partners to the internal actin cytoskeleton at this point and may provide at least a partial explanation for why AMIS localisation occurs precisely at this region. We therefore suggest that the most relevant information to our study would be to determine whether either JAM or Nectin proteins are specifically localised at the AMIS, alongside PAR-3 and ZO-1, and might therefore be somehow enabling this stabilisation of E-Cadherin. We therefore plan to carry out IHC stains for JAM-A (new name for JAM-1), which has been found to be present in the mouse inner cell mass, to determine where it is localised within the mESC cell clusters with/without cell division and in WT/*Cdh1* KO cells. We will update the supplementary results and discussion accordingly in the final manuscript.

Depending on these results, we might also try to knock down the function of JAM-A, using siRNA. If successful knock down were achieved, we would carry out FRAP to determine whether E-cadherin junctional stability had been altered and would also stain for AMIS markers such as PAR-3 and determine whether Golgi and centrosomes were polarised. However, it is important to note that, although we were able to achieve E-cadherin RNAi to a certain degree, it is not always possible to achieve sufficient knock down of protein by the 24-hour AMIS

Revision Plan

timepoint. Since the results of these experiments would not alter the impact of our pre-existing data, we do not propose to create new knock out cell lines in the current study. Also, possible redundancy between different paralogs may affect the interpretation of this experiment so we would only include these results if they allowed for clear interpretation.

A previous study (Gao L ,et al. Development. 2017) has already shown that knocking out Afadin (which would therefore disable Nectin junctions) in MDCK cell 3D cultures did not affect initial AMIS formation or localisation, although later cell division orientation and therefore lumen positioning was affected. Afadin was also not localised to the AMIS. Therefore, it is less likely that Nectin is involved in AMIS localisation and while we will stain for its localisation by IHC, we don't propose to try to knock down its function.

Reply to Reviewer 2 Point 1

The reviewer pointed out using a different mitosis blocker beside Mitomycin C. a) In the updated manuscript, we included one additional drug treatment: Aphidicolin. The results showed the AMIS could form in the centre of cell-cell contacts in Aphidicolin treated, division-blocked cells. AMIS (PAR3, ZO1) and the Golgi network was also polarised towards this point (Figure S1 G-I). In the final manuscript, we will include a full data set with N=3 independent experiments. Though the same as Mitomycin C, Aphidicolin is a DNA replication blocker, it confirmed that the AMIS formation upon treatments is not a Mitomycin-only artefact. b) As the reviewer suggested to block mitosis at the M phase, we are testing using microtubule polymerization inhibitors, Nocodazole and Taxol and will include these results if appropriate. However, these treatments will also affect the cytoskeleton, significantly affecting the cell shape and potentially interrupting the cell-cell contact interface. Therefore, it may not be possible to include these experiments.

Reply to Reviewer 2 Point 2

The reviewer suggested to include more examples of movies showing 2 and 4 cell cluster formation in division blocked conditions. We will be happy to provide more examples of the movies included in Figure 2 and Movie 2 in the final submission. The puncta in submitted Movie 2 was not as clear as the in Figure 2D as the reviewer pointed out. This was largely due to the reduce-sized movie in the original submission. We will provide full-resolution movies in the final submission. We do often see the 'perfect' 4-cell shape in division-blocked cells (e.g. the last frame of movie 2, shown at timepoint 19:00 in figure 2D). The shape of the clusters appears largely dependent on how many cells fuse together.

Reply to Reviewer 2 Point 4

The reviewer reasoned that the WT and Cdh1 KO mESC were from different genetic backgrounds. The WT (ES-E14) mESCs were generated from 129P2/Ola mice and the Cdh1 KO mESCs were generated from

Revision Plan

129S6/SvEvTacArc mice. To confirm the results acquired based on the two cells lines, we are doing two approaches: 1) As the reviewer suggested, we are using siRNA knock-down of E-cadherin in the Wild-type mESCs (ES-E14) to confirm the results we had of the AMIS absence in the E-cadherin knock-out mESC cultures. As Figure S2C,D now shows, the concentrated PAR3 between two mESCs was largely reduced after E-cadherin knock-down. We will also include Mitomycin-treated conditions in this experiment for the final publication. 2) As an alternative approach, not dependent on RNAi functionality, we have acquired a 129S6/SvEvTacArc background mESC (the W4 line) as the wild-type mESC line that has the same background as the *Cdh1* KO mESC line. We are using this line to perform the control experiments of Figure 3A-C to confirm the previous results, which so far are comparable in both the ES-14 and W4 mESC cell lines. Our preliminary data below show the same results as we had with the ES-E14 cells in the current Figure 3A. We will finish the full data set of N = 3 experiments and replace the current Figure 3A-C, S2A data with that from the W4 mESC cell line. In the meanwhile, we have labelled the type of wide type mESC used for each experiment in the manuscript.

Reply to Reviewer 3 Point 1

The reviewer pointed out we should include three independent experiments for our data in Figure 4E. We agree with the reviewer. We are very happy to do the suggested experiments and data analysis and will be able to provide the data of N=3 independent experiments in the final manuscript.

Reply to reviewers' minor points

We will provide a co-staining of mCherry (to label mCherry-PAR6B), Phalloidin and PAR-3 in a more updated manuscript to replace Figure 2A.

3. Description of the revisions that have already been incorporated in the transferred manuscript

Reply to Reviewer 1 Point 1

The reviewer pointed out a possibility that the Golgi polarisation leads to local/centre-most regional E-cadherin junction “maturation”, then contribute to AMIS seeding. To address this suggestion, we did fluorescence recovery after photobleaching (FRAP) using a mESC line that expresses E-cadherin-GFP in the updated manuscript. We compared the recovery speed and rate in the centre-most region and side regions to discuss whether E-cadherin junctions have different stability at these regions. What we found is that though the E-cadherin and E-cadherin-GFP protein level is at the same level at the two regions in mESC doublets (Figure S3), the mobile fraction of E-cadherin-GFP is lower in the centre-most region than the side regions (Figure 3 I, J). This implies that E-cadherin junctions in the centre-most region are more stable. We have included corresponding description of this data in Results, Methods and Discussion. We will also include equivalent data from non-mitomycin c treated control cells in the final manuscript.

Still, we do not know whether the more stable E-cadherin junctions were due to the Golgi polarization, but we have included the possibility of Golgi polarisation leads to local E-cadherin maturation in our Discussion in the transferred manuscript as follows:

“In addition, a recent study of chick neural tube polarisation (where N-Cadherin is the dominant Cadherin) has demonstrated that the interaction of β -catenin with pro-N-cadherin in the Golgi apparatus is necessary for the maturation of N-Cadherin, which is in turn important for apicobasal polarity establishment (Herrera et al, 2021). This provides the possibility that the polarised Golgi apparatus that we observe in the mESC clusters might be directionally delivering mature E-cadherin to the central-most region of cell-cell contact.”

Reply to Reviewer 2 Points 3 & 5 and Reviewer 3 Point 2

We appreciate the comments from the reviewers regarding qualifying some of the discussion of our results.

Reviewer 2 points out that E-cadherin is not providing a ‘Symmetry breaking’ step, since cells are eventually able to polarise in the absence of E-cadherin (even though they can’t make an AMIS). We have therefore modified our discussion of this point to read: “Our results therefore suggest that Cadherin-mediated cell-cell adhesion may provide the spatial cue required for AMIS localisation during *de novo* polarisation.”. The last paragraph of the manuscript now reads: “In summary, our work suggests that Cadherin-mediated cell-cell adhesion is necessary for localising the AMIS during *de novo* polarisation of epithelial tubes and cavities.”

Reviewer 3 points out that the E-Cadherin molecule by itself is not sufficient to recruit the AMIS proteins to the centre-most region of the cell-cell contacts since E-cadherin is localised all along the cell-cell contact. We have now included a FRAP analysis demonstrating that E-cadherin is more stable in the centre-most region of cell-cell contacts (Figure 3I,J), which supports the role of E-Cadherin in directing AMIS localisation to this centre-most region. Nevertheless, we accept the reviewer's point that we still do not know the downstream mechanism by which the AMIS is precisely localised to the central region of cell-cell contacts, and we have extended our discussion of this point in the updated manuscript. To clarify the language, we have also altered our results heading and other references to this point to read: "E-Cadherin adhesions are sufficient to **initiate** AMIS localisation, independent of ECM signalling and cell division". We believe our experiments with two methods support this claim that the formation of E-cadherin-based adhesions without cell divisions and ECM signals are sufficient to initiate AMIS localisation; in particular Figure 4C-E, in which a centrally located AMIS formed even in a reductionist system of only 1 cell plated on E-cadherin covered glass.

Reply to Reviewers' minor points

We have revised our texts, made the nomenclature of protein PAR3 consistent, and included the information of antibody suppliers, as the reviewers pointed out. Specific response to Reviewer 2; in p2 and p7, the texts were referring to zebrafish studies, where PAR3 is referred to as Pard3. We have marked it with "Pard3 (PAR-3)" now. We have increased the size of images in figure 5B and inverted the colour to make it more visible. Since this made the figure too big, we moved the ZO1 images to Figure S5A.

4. Description of analyses that authors prefer not to carry out

Reply to Review 3 Point 3

We agree with the reviewer. Our current data set of live imaging at day 3 are used to confirm the idea from the fixed images that a wrapping process does happen for lumenogenesis during the *Cdh1* KO cyst formation. The current dataset could not exclude the possibility that the hollowing might co-exist. The reviewer therefore suggests including a live movie depicting early stages (before 78:00) of E-Cadherin knock-out cluster development. We did try to collect this data before we first submitted the manuscript but encountered significant technical problems due to the high sensitivity of early stage *Cdh1* KO cells to phototoxicity. This meant that we could not image with less than one hour interval nor over longer than 24 hour and were therefore unable to analyse how the cell clusters behave before forming the cup-shaped cavity. We will attempt these experiments again (e.g. imaging from 12-24 hours and 24-36 hours). However, there is a high likelihood that the experiments will not be technically possible, which is why we list them in section 4 of our review plan. Instead, we include the following sentence in our discussion: "We were

Revision Plan

unable to live-image earlier stages of *Cdh1* KO cluster development due to the sensitivity of these cells to phototoxicity so we can't exclude the possibility that hollowing lumenogenesis occurs in parallel, although our IHC analysis does not indicate that this is the case."

Dear Dr. Buckley,

Thank you for transferring your manuscript with its Review Commons referee reports and responses to The EMBO Journal.

Given the referees' positive recommendations, I would like to invite you to submit a revised version of the manuscript, following your submitted revision plan. I should add that it is EMBO Journal policy to allow only a single round of revision, and acceptance of your manuscript will therefore depend on the completeness of your responses in this revised version. If you would like to discuss anything about the manuscript or publication process, I can offer to arrange a Zoom chat.

Thank you for the opportunity to consider your work for publication. I look forward to your revision.

Yours sincerely,

William Teale

William Teale, PhD
Editor
The EMBO Journal
w.teale@embojournal.org

We realize that it is difficult to revise to a specific deadline. In the interest of protecting the conceptual advance provided by the work, we recommend a revision within 3 months (7th Jun 2022). Please discuss the revision progress ahead of this time with the editor if you require more time to complete the revisions. Use the link below to submit your revision:

We would like to thank all the reviewers for their time and for their positive and constructive review of our study. We are happy that they all regard this as a highly significant piece of work. We have addressed their suggestions in our updated manuscript and outlined these changes in our responses below.

Reviewer #1

Reviewer #1 (Evidence, reproducibility and clarity (Required)):

Summary:

Formation of tubes in a developing organism may arise from the closure of a pre-existing polarized epithelium or from de novo polarization and cavity formation in group of dividing cells. The concept of apical membrane initiation site (AMIS) refers to the fact that polarity proteins as PAR3 accumulate at a point where the apical membrane will be created. This accumulation occurs as early as the two-cell stage. Previous reports have demonstrated the importance of the division process in defining this AMIS, however, in the present work the authors in vitro 3D cultures of mESC to report a mitosis independent mechanism that creates an AMIS, induces the polarization of groups of two or more cells, and permits the formation of a central cavity. The report shows that the mechanism is fully dependent on the polarized accumulation of E-cadherin at the cell membrane in contact with the other cells. Moreover, the mechanism does not require mitosis or interaction with the extracellular matrix.

Major comments:

The main objective of the work is to demonstrate that AMIS creation and cavity formation can be mitosis independent and that it is dependent on the accumulation of E-cadherin at the midline between two cells in contact. To demonstrate these objectives, the authors perform 3D cultures of mESC. To rule out the requirement of mitosis the authors perform cultures that are treated with mitomycin C and the purify single cells that are cultured again. The authors show time-laps experiments demonstrating that individual cells that do not divide create an AMIS when they contact one to each other. With this cultures they demonstrate that the process does not require an interaction with ECM (provided by the matrigel) but requires E-cadherin, to demonstrate, that they use E-cadherin KO cells (the same line where E-cadherin has been deleted). The work is well written and the objectives very clear. The technology used and the experiments done are adequate and sufficient to accomplish the proposed objectives and the results obtained clearly support the conclusions reached. The methods are well explained and transparent to be reproduced elsewhere and the number of replicas and the statistical methods applied seem correct to me, although I am just a biologist, not a mathematician.

Although the objectives of the work, that are: to demonstrate that AMIS formation can be independent of mitosis and that AMIS requires E-cadherin, there are parts of the results that could be further studied or at least discussed more thoroughly.

1) Firstly, the authors show that in non-dividing cells an AMIS is formed at the first contact site between the two cells, they also show that in the absence of E-cadherin the cell maintains the polarization of centrioles and Golgi apparatus, in spite that no AMIS is formed, this indicates that the deposition of E-cadherin at the midline membrane is part of a more global polarization event that most likely is initiated by the a directional activity of the Golgi apparatus that may direct the delivery of mature E-cadherin in that particular direction, initiating or maintaining the basis for an AMIS, since recent work (already cited in the manuscript) has demonstrated the importance of cadherin maturation for polarity establishment and maintenance (Herrera et al, 2021), the actual results should be further discussed in this context.

Reply to the comment:

The reviewer pointed out a possibility that the Golgi polarisation leads to local/centre-most regional E-cadherin junction "maturation", then contribute to AMIS seeding. To address this suggestion, we did fluorescence recovery after photobleaching (FRAP) using a mESC line that expresses E-cadherin-GFP in the updated manuscript. We compared the recovery rate in the centre-most region and side regions to determine whether E-cadherin adhesions have different stability at these regions. What we found is that although the E-cadherin and E-cadherin-GFP protein level is at the same level at the two regions in mESC doublets (Figure EV3), the mobile fraction of E-cadherin-GFP is lower in the centre-most region than the side regions (Figure 3 J, K). This implies that E-cadherin junctions in the centre-most region are more stable. We have included corresponding description of this data in Results, Methods and Discussion. Still, we do not know whether the more stable E-cadherin junctions were due to the Golgi polarization, but we have included the possibility of Golgi polarisation leads to local E-cadherin maturation in our Discussion as follows:

"In addition, a recent study of chick neural tube polarisation (where N-Cadherin is the dominant Cadherin) has demonstrated that the interaction of β -catenin with pro-N-cadherin in the Golgi apparatus is necessary for the

maturation of N-Cadherin, which is in turn important for apicobasal polarity establishment (Herrera et al, 2021). This provides the possibility that the polarised Golgi apparatus that we observe in the mESC clusters might be directionally delivering mature E-cadherin to the central-most region of cell-cell contact.”

2) Secondly, it was previously shown that in different epithelia, upon cell-cell contact, the aPKC complex (that includes Par3 and Par6) is recruited early to the contact site where with the participation of Cdc42, aPKC is activated generating an initial spot-like adherent junction (AJs) (Suzuki et al., 2002). In that case it is thought to be mediated by a direct interaction between the first PDZ domain of PAR-3 and the C-terminal PDZdomain-binding sequences of immunoglobulin-like cell adhesion molecules: JAM-1 and nectin-1/3 (Fig. 3) (Ebnet et al., 2001; Itoh et al., 2001; Takekuni et al., 2003). Thus it would be interesting to know if AMIS formation in absence of cell division depends on JAM-1 or nectin and whether JAM-/Nectin signalling is sufficient to initiate the Golgi and centriole polarization and which is the mechanism governing it.

Reply to the comment:

The reviewer suggests it would be interesting to know whether there is a role for the proteins JAM-1 or Nectin in AMIS formation. Like E-Cadherin, these are transmembrane junctional proteins that are present at the initiation of spot adhesions in epithelial 2D monolayers and are known to be part of a complex network of interactions between PAR-complex, junctional molecules, MAGUK scaffolding proteins and the actin cytoskeleton. In Figure 4 and EV4 of the revised manuscript, we used immunofluorescence to investigate the localisation of these proteins at AMIS stages. We also used RNAi to knock down JAM-A and Nectin-2, the forms of JAM and Nectin that were reported to be expressed between inner cell mass cells in the mouse blastocyst. Our results showed that JAM-A and Nectin-2 were localised at the AMIS later than PAR-3 and that, even upon a sufficient knock-down of these adhesion molecules, PAR-3 polarisation was not affected. Therefore, we conclude that JAM-A and Nectin-2 are not necessary for AMIS localisation. We have added a new results section detailing these findings, alongside an investigation of P-cadherin, which also was not necessary for AMIS localisation. In addition, it is important to note that our results already demonstrate that E-cadherin-based adhesions are sufficient to **initiate** AMIS localisation. For example, our results from Figure 5C-F demonstrate that, in a reductionist system of a single cell plated on E-cadherin covered glass, a centrally located AMIS still forms. Precisely unravelling the mechanisms by which this happens would be better for a future study (which we have now stated in the Discussion).

Minor comments:

As I mentioned before, the paper is well presented and very clear, yes it is simple, but simple is always better, no complicated graphics or letterings, thank you. Although in my opinion the work is very well written, I have to admit that I am not qualified to evaluate the literary style of the work since English is not my mother tongue, also I have not reviewed typographical errors since I think that is the work of the editorial, not of scientific reviewers.

Please include the full reference of all the antibodies used, including the company and not just the catalog number.

Reply to the comment:

We have now included full details of all the antibodies used in the methods section.

Reviewer #1 (Significance (Required)):

The paper describes for the first time that contrary to what was previously believed an AMIS can be generated without a cell division. This is very important because it opens the possibility that the mechanisms that originate the biologic cavities are in fact not really how we believed.

The work is of interest of all cell biology scientists, specially working in developmental biology, cancer research. My particular field of expertise is cell biology and signaling, always applied to particular events as nervous system development or cancer, in particular I am interested in Wnt/b-catenin and Sonic Hedgehog pathways.

Reply to the comment:

Thank you for your helpful and constructive comments.

Reviewer #2

Reviewer #2 (Evidence, reproducibility and clarity (Required)):

Summary:

The importance of cell division and the post-mitotic midbody in the establishment of the apical membrane initiation site (AMIS) is quite well established. However, there are observations hinting to a cell division-independent mechanism of the AMIS formation. The authors hypothesized that cell adhesion involving E-cadherin could direct the site for AMIS localisation during de novo polarisation.

As model system the authors used mouse embryo stem cell (mESC) culture in Matrigel, which has been used as an in vitro model for the de novo polarisation of the mouse epiblast. The slow lumen formation in culture allows for a relatively clear separation of the stages of de novo polarisation. This enables to study the initiation of apico-basal polarity of embryonic cells alongside the first cell-cell contacts between isolated cells and small cell clusters. Here, the goal was to determine the role of cell adhesion, and in particular E-cadherin, in mESC AMIS localisation.

Major comments:

1) Mitomycin C is commonly used to block cell division, however, what it does is it blocks DNA replication, and the blockage of cell division is a consequence. It could have other effects than only blocking cell division. What about using a mitosis blocker? It would be good to have a second way in addition to mitomycin C treatment of confirming that the results support the conclusion that cell division is dispensable for AMIS localization, as the full work builds up on that first observation and this experimental setup is carried on through the manuscript.

Reply to the comment:

In the updated manuscript, we included one additional drug treatment: Aphidicolin. The results showed the AMIS still forms in the centre of cell-cell contacts in Aphidicolin treated, division-blocked cells. AMIS (PAR3, ZO1) and the Golgi network was also polarised towards this point (Figure EV1 D-H). Although Aphidicolin is also a DNA replication blocker, these results confirmed that AMIS formation still occurs in division-blocked cells, using two different treatments. The reviewer suggested using a mitosis blocker. However, these inhibitors (such as nocodazole and Taxol) all affect microtubule polymerisation. It is known that the microtubule cytoskeleton is necessary for PAR-3 localisation at the AMIS. For example, in a previous publication, we treated zebrafish neural rods with nocodazole, which reversibly displaced PAR-3 from the middle of the organ primordium to the outer end of neuroepithelial cells: doi:10.1038/emboj.2012.305). Therefore, treating mESCs with drugs that affect microtubule polymerisation would likely prevent PAR-3 localisation, but this would likely be due to a lack of an in-tact microtubule cytoskeleton. Disrupting microtubules may also disrupt the cell-cell contact interface, further complicating interpretation of these results. Therefore, we have not included these experiments in the manuscript. Since we found that blocking division does **not** result in a change in PAR-3 localisation (as opposed to it causing a loss in localisation), we don't feel that our results show any off-target effects of blocking DNA replication but instead robustly demonstrate that blocking cell division does not prevent AMIS localisation.

2) Many clusters at higher cell stage that are shown (e.g. Fig 2C, 3A), look like they are in the perfect 4-cell stage after cell division. Movie 1 does not show that, only 2 cells are clustering. Movie 2 shows how two 2-cell clusters form a 4-cell cluster, however, that does not look as "perfect" as the 4-cell stages shown in the figures, which look as said more like 4-cell stages resulting from cell division. Maybe the authors could provide more movies that show the 2-cell cluster doublets (4-cell clusters)? Also, the puncta relocalization to the cell-cell contacts in the 2-cell cluster doublet is not so very clear in the movie 2. Maybe the authors have more movies for 2-cell cluster doublets?

Reply to the comment:

We think that the Referee is referring to the shape of 4-cell clusters formed in the absence of division (when treated with mitomycin).

Movie EV1 panel D shows 4 individual division-blocked cells that then merge together to make a cell cluster. The end resolution is not good enough to show the individual cells making up the cluster, but these can be clearly seen at the start of the movie. The orientation of the individual cells making up the cluster is better seen in other figures from the manuscript where immunofluorescence is used (such as figure 2B or 3A), which are representative of the data seen when 4 cells coalesce.

Movie EV2 panel D shows the two 2-cell clusters forming a 4-cell cluster, as the referee points out. Images from the same movie are shown in Figure 2D. The last frame of the movie shows a relatively 'perfect' 4-cell stage, shown at timepoint 19:00 in figure 2D. The previously submitted Movie 2 was not as clear as in Figure 2D as the reviewer pointed out. This was largely due to the reduce-sized movie in the original submission. We now include a full-resolution Movie EV2 and so hope that this clarifies the issue. We also now include some time frames taken from additional movies, to show the relocalisation of mChy-PAR6B to regions of cell-cell contact (Fig S2D,E).

3) On page 4 the authors observe: "Whilst homogenous control doublets localised PAR-3 to the central region of the cell-cell interface, heterogeneous chimeric doublets did not localise PAR-3 centrally (Figure 3D,E). Golgi and centrosome localisation towards the cell-cell interface suggested that the overall axis of polarity was maintained, even in the absence of both cell division and E-CADHERIN (Figure 3F-H & S2C)." This means that the axis of polarity is established without E-cadherin and without the AMIS. So, the cell-cell contact site itself is able to establish polarity, but not localize the AMIS.

In line with this, the authors state: "The results also demonstrate that ECM in the absence of E-CADHERIN is insufficient for AMIS localisation." So, without E-cadherin, there is no AMIS, the ECM cannot establish it, but polarity is still established. On p. 6, it is then concluded that E-cadherin and centralised AMIS localisation are not required for apical membrane formation, but that they promote its formation earlier and more efficiently in development. In the discussion, however, the authors then state in a rather contrary manner "Our results therefore suggest that CADHERIN-mediated cell-cell adhesion may provide the symmetry-breaking step required for AMIS localisation during *de novo* polarisation." However, cells are polarized and have an apical membrane (Figure 3F-H & S2C) without cadherin and the AMIS, so how can this be the symmetry-breaking step for *de novo* polarization? Later on, in line with the earlier statement that E-cadherin and the AMIS location are not required for apical membrane formation, the authors then refine the previous statement: "the role of E-CADHERIN in *de novo* polarisation is specifically to localise the AMIS, which enables the integration of individual cell apical domains to a centralised region preceding lumen hollowing." This seems to be more likely to me than the CADHERIN-mediated cell-cell adhesion as the symmetry-breaking step. I therefore disagree in this point with their overall summary on p. 8, and find this a bit confusing.

Reply to the comment:

The reviewer pointed out that E-cadherin is not providing a 'symmetry breaking' step, since cells are able to polarise in the absence of E-cadherin (even though they can't make an AMIS). We thank the reviewer for pointing out this confusion. We have modified our discussion of this point to read: "Our results therefore suggest that Cadherin-based cell-cell adhesion may provide the spatial cue required for AMIS localisation during *de novo* polarisation.". The last paragraph of the manuscript now starts: "In summary, our work suggests that Cadherin-mediated cell-cell adhesion directs AMIS localisation during *de novo* polarisation of epithelial tubes and cavities."

In addition, we have removed the mention of symmetry-breaking from the introduction and only include it once, in the discussion, in reference to the published suggestion that the formation of the midbody provides a symmetry-breaking event in AMIS localisation: "Rather than acting as the initial symmetry-breaking step in AMIS localisation, we suggest that tethering of apically directed proteins to the midbody might instead act to transiently align cell division, cell adhesion and the forming apical domain, therefore enabling an organised structure to be generated in the presence of dynamic cell movement and tissue growth".

4) As Wild-type mESCs (ES-E14) were purchased from Cambridge Stem Cell Institute and Cdh1 KO mESCs were gifted from a lab, it would be good to (genetically) characterize the cell lines because, apparently, they do not have the same origin and the KO cells were not derived from the parental mESCs. Alternatively, a control experiment with knockdown of Cdh1 in the purchased mESCs could be done, even if that would not lead to a complete knockdown, to make sure that the observed effects are the same as with the Cdh1 KO cell line.

Reply to the comment:

The reviewer reasoned that the WT and Cdh1 KO mESC were from different genetic backgrounds. The WT (ES-E14) mESCs were generated from 129P2/Ola mice and the Cdh1 KO mESCs were generated from 129S6/SvEvTacArc mice. To confirm the results acquired based on the two cells lines, we have done two approaches:

1) As the reviewer suggested, we used siRNA knock-down of E-cadherin in the Wild-type mESCs (ES-E14) to confirm the results we had of the AMIS absence in the E-cadherin knock-out mESC cultures. As Figure S3 now shows, PAR-3 did not localise centrally between two mESCs after E-cadherin knock-down, in both control and in mitomycin C treated, division-blocked cells.

2) As an alternative approach, not dependent on RNAi functionality, we acquired a 129S6/SvEvTacArc background mESC (the W4 line), which is the wild-type mESC line that has the same background as the *Cdh1* KO mESC line. We used this line to perform the control experiments of Figure 3A-C. We found that results were comparable in both the ES-14 and W4 mESC cell lines and have replaced our previous Figure 3A-C with the new data from the W4 mESC cell line in the revised manuscript.

5) The existing data is carefully analyzed with appropriate statistics. Replicates are sufficient. The conclusions are not yet fully justified, as discussed.

Reply to the comment:

We feel that we have now fully justified our conclusions, as detailed in our responses above.

Minor comments:

- Fig S1D: It is labeled "Pard3". While also correct, it should be consistent, i.e. PAR3.
 - P. 5 remove (slightly)
 - P. 2, p. 7 "Pard3", replace with PAR3
 - The antibody lists already contain catalog numbers in case of the primary antibodies, but no suppliers. They should be added, and also specified for the secondary antibodies for better reproducibility.
- In general, the text and figures are well prepared and of high quality. The citations appear appropriate.

Reply to the comment:

We have now included full details of all the antibodies used in the methods section. We have revised our texts and labelled PAR-3 consistently. In p2 and p7, the texts were referring to zebrafish studies, in which species PAR-3 is referred to as Pard3. We have now marked it with "Pard3 (PAR-3)".

Reviewer #2 (Significance (Required)):

Comment 1

The work describes the conceptual novelty of cell adhesion as alternative mechanism to cell division for AMIS localization, and in particular E-Cadherin as being required for AMIS positioning. It is still unclear why the AMIS is centered and the localization of cadherin is equal along cell-cell contacts (Fig 2C, S1E). How do Cadherin localization dynamics look like during the clustering of two cells? During cell division in a MDCK cyst (which is where my expertise lies), cell adhesion has to be partially removed during cytokinesis and abscission, and then be installed again, basically like a new cell-cell contact. Thus, could E-cadherin focus ("trap") the "AMIS initiation seed", rather than direct binding of PAR3 /PAR6 to cadherin as discussed by the authors, since E-cadherin is localized along the whole contact site and not centred? Could the unknown "apical seed" (which in cell division is the midbody) be trapped by cell adhesion? Could this be a common mechanism between cell division- and cell adhesion-driven AMIS localization? This finding could therefore have an even broader impact. What are the author's thoughts? While my speculation might be wrong, it might be worth hypothesizing on the connection between the role of E-cadherin in the two ways of AMIS localization.

Reply to the comment:

Thank you for this interesting suggestion. Our new FRAP results show that, though the E-cadherin and E-cadherin-GFP protein level is at the same level along the whole cell-cell interface (Figure EV3), the mobile fraction of E-cadherin-GFP is lower in the centre-most region than the side regions (Figure 3 J, K). This implies that E-cadherin junctions in the centre-most region are more stable, both in post-mitotic control cells and in division-blocked cells that have adhered together. This might suggest that E-cadherin is more stably bound via its downstream partners to the internal actin cytoskeleton at this point, which might help to stabilise or to localise AMIS proteins. However, we still don't know the full mechanism by which AMIS proteins localise to the central-most point of cell-cell contact we agree that a separate study on the dynamics of E-cadherin during clustering would be very interesting in the future. We have now discussed this point further in the discussion section and have stated that "Uncovering the mechanisms directing adhesion-dependent AMIS localisation precisely to the midpoint of cell-cell adhesions will be an interesting area for future studies".

Comment 2

Another novelty is the observation that polarity and cavities form later on in development independently of E-cadherin and an AMIS. This type of mechanism should be discussed further and put more into perspective with the literature.

Reply to the comment:

We thank the reviewer for the comment. We have now included the following sentence in our discussion:
“Therefore, *Cdh1* KO cells do appear to still make an apical membrane (presumably directed by ECM-mediated signalling) but do so more slowly than in WT cells and without going through a centralised AMIS stage.”

We now also discuss the potential role of other adhesion molecules during later polarisation and lumenogenesis:
“Although the other adhesion molecules we have tested (P-cadherin, JAM-A and Nectin-2) did not contribute to centralised AMIS formation, mESCs cultured in Matrigel and mouse inner cell mass cells only become fully epithelialised and start to generate the central cavity once they have exited pluripotency and there are multiple cells in the structures (Kim et al., 2021; Shahbazi et al., 2017). Thus, whilst E-cadherin appears to be essential for AMIS localisation, other adhesion molecules may be important at later polarisation and lumenogenesis stages.”

Comment 3:

The work describes a new mechanism which could be of broad importance in developmental biology. I therefore think that this work is highly significant.

Reply to the comment:

Thank you for your helpful and constructive comments.

Reviewer #3

Reviewer #3 (Evidence, reproducibility and clarity (Required)):

In this manuscript, Xuan Liang and collaborators shed light on how the precise localisation of the apical membrane initiation site (AMIS), necessary for organised lumen formation, is directed at the single-cell level. By characterising de novo polarising mouse embryonic stem cells (mESCs) cultured in 3D, the authors have uncovered a division-independent mechanism of de novo polarisation and AMIS localisation based on adhesion molecules. More precisely, they suggest that E-CADHERIN-mediated cell-cell adhesion may provide the symmetry-breaking step required for AMIS localisation during de novo polarisation since this molecule alone is sufficient and necessary to drive correct AMIS localisation. Interestingly, a high proportion of E-Cadherin knock-out (Cdh1 KO) mESC cell clusters do not hollow but instead generate lumen-like cavities via a closure mechanism. Despite not knowing the mechanism involved in the closure of these lumen-like cavities, the role of E-CADHERIN in de novo polarisation would be associated with initial steps in lumen formation (AMIS formation and localisation) but not in later steps where E-Cadherin knock-out mESC cell clusters can still make an apical membrane but do so more slowly than in WT cells and without going through a centralised AMIS stage.

Altogether, this study supports their previously published zebrafish neuroepithelial cell in vivo analysis, which demonstrated the division-independent localisation of Pard3 and ZO-1 at the neural rod primordial midline (Buckley et al., 2013). The authors have provided a novel mechanism of de novo polarisation and AMIS formation that occurs in vivo and in vitro. For this reason, this is a work with great significance that will undoubtedly be of general interest to the readers of Review commons. Nonetheless, several issues should be addressed before the publication of this manuscript.

Major points:

1) In figure 4, the authors tried to demonstrate that E-CADHERIN is sufficient for AMIS localisation, independent of ECM signalling and cell division. To this end, they cultured individual division-blocked mESCs onto either E-CADHERIN-FC recombinant protein or FIBRONECTIN pre-coated glass and carried out IHC for PAR-3 after 24 hours in culture. They then performed heatmaps and analysed PAR3 intensity (Fig. 4 D, E). Although the data presented are fascinating and show the effects the authors describe, the authors should improve their sample number and repeat this experimental procedure and analysis at least two more times for their results to be consistent (only 15 cells in one experiment have been used to carry out the statistical analysis described previously).

Reply to the comment:

We agree with the reviewer and have now provided n=3 independent experiments in the revised manuscript (now Figure 5 D-F). Both individual cell values (small dots) and mean experimental values (large dots) are now shown in our quantifications so that all the information on experimental replicates is included.

2) The expression of E-cad is necessary for the proteins that define the apical membrane (Par3, Par6, aPKC) to be located in the AMIS. The results are clear and robust. Even so, I do not think this is sufficient, as the authors claim (headline of page 5, Figure 4). It seems clear that something more than Ecad is needed for the localisation of Par3 in the AMIS because, as the authors indicate in the discussion, Ecad is located along the entire cell-cell junction, while par3 is focused on AMIS. There must be something else that is necessary for the location of Par3. Therefore, the experiment in figure 4 does not prove that E-cad is sufficient but confirms that it is necessary for that location. Another series of experiments would have to be carried out to prove that it is sufficient. This must be clearly stated in the final version of the manuscript.

Reply to the comment:

The reviewer pointed out that the E-Cadherin molecule by itself is not sufficient to recruit the AMIS proteins to the centre-most region of the cell-cell contacts since E-cadherin is localised all along the cell-cell contact. We have now included a FRAP analysis demonstrating that E-cadherin is more stable in the centre-most region of cell-cell contacts (Figure 3J, K), which supports the role of E-Cadherin in directing AMIS localisation to this centre-most region. Nevertheless, we accept the reviewer's point that we still do not know the downstream mechanism by which the AMIS is precisely localised to the central region of cell-cell contacts, and we have extended our discussion of this point in the updated manuscript. To clarify the language, we have also altered our results heading and other references to this point to read: "E-Cadherin adhesions are sufficient to **initiate** AMIS localisation, independent of ECM signalling and cell division". We believe our experiments with two methods support this claim that the formation of E-cadherin-based adhesions without cell divisions and ECM signals are sufficient to initiate AMIS localisation; in particular, Figure

5C-F, in which a centrally located AMIS formed even in a reductionist system of only 1 cell plated on E-cadherin covered glass.

3) It is clear that E-Cadherin knock-out mESC cell clusters open cup-shaped cavities before generating a lumen-like structure. Fig. 5 presents compelling data about this in vivo lumen formation mechanism without hollowing, though they briefly describe this process. Whilst I am conscious that they do not know the mechanism by which such 'closure' occurs and that this would be suitable for another manuscript, I would strongly suggest including a live movie depicting early stages (before 78:00) of E-Cadherin knock-out cluster development. Many queries arise with this piece of data, as it seems that a small lumen could be forming prior to the cup-shaped cavity.

Reply to the comment:

Our original data set of live imaging at day 3 was used to confirm the idea from the fixed images that a wrapping process does happen for lumenogenesis during the *Cdh1* KO cyst formation. However, that dataset could not exclude the possibility that the hollowing might co-exist. The reviewer therefore suggested including live movies depicting early stages (before 78:00) of E-Cadherin knock-out cluster development.

We did try to collect this data with movies that lasted from day 2 till day 4 before we first submitted the manuscript. However, we encountered significant technical problems due to the high sensitivity of early stage *Cdh1* KO cells to phototoxicity. This meant that we could not image for longer than 24 hours and were therefore unable to analyse how the cell clusters behaved before forming the cup-shaped cavity. To overcome these technical challenges, we have instead captured shorter movies of earlier stage mESC clusters starting from day 2 in culture, in addition to our later movies starting from day 3 in culture. This is detailed in the results section as follows:

“This confirmed that, whilst the WT cell clusters made a central lumen (8/8 movies on day 2) and then expanded this already central lumen (8/8 movies on day 3), *cdh1* KO cell clusters first generated an open cup-shape cavity (3/3 movies on day 2), which then gradually closed, eventually generating a centralised lumen-like structure without hollowing at a later stage of development (3/3 movies on day 3).”

We now also include the following sentences in our discussion of this point:

“Our movies of *Cdh1* KO cell clusters (Movies EV3 & 4) confirmed conclusions from fixed data (Fig 6) that *Cdh1* KO cell clusters first generate a polarised, open cup-shape cavity, before ‘closing’. Due to phototoxicity, we only had limited sample size and movie lengths, thus we were not able to fully exclude the possibility that the hollowing lumenogenesis occurs to some small extent in parallel, but our data is not suggestive of hollowing lumenogenesis in the *Cdh1* KO cell clusters.”

Minor points:

1) Fig. 2A: Actin staining could be included to better visualise the spheroids.

Reply to the comment:

We have now included the brightfield channel in Figure S2A to better visualise the cell clusters in Figure 2A. We have also included a co-staining of mCherry-PARD6B, PAR-3 and F-actin in Figure S2B.

2) Fig. 5B is very small, I would recommend them to present it bigger.

Reply to the comment:

We have increased the size of images (now in figure 6B) and inverted the colour to make it more visible. Since this made the figure too big, we moved the ZO-1 images to Figure EV5.

3) I would encourage the authors to revise the figures as some have displaced text. (see Fig. 5, 72 hours, *Cdh1* KO).

Reply to the comment:

We have revised our figures and corrected these issues.

Reviewer #3 (Significance (Required)):

Altogether, this study supports their previously published zebrafish neuroepithelial cell in vivo analysis, which

demonstrated the division-independent localisation of Pard3 and ZO-1 at the neural rod primordial midline (Buckley et al., 2013). The authors have provided a novel mechanism of de novo polarisation and AMIS formation that occurs in vivo and in vitro. For this reason, this is a work with great significance that will undoubtedly be of general interest to the readers of Review commons. Nonetheless, several issues should be addressed before the publication of this manuscript.

My lab work in lumen formation in 3D organotypic cultures and organoids.

Reply to the comment:

Thank you for your helpful and constructive comments. We feel that we have now fully addressed the issues raised, as detailed in our responses above.

Dear Clare,

We have now received re-review reports from two referees. As you will see, you have addressed their concerns satisfactorily. Before moving towards publication, there are some remaining editorial points which need to be addressed. In this regard would you please:

- update the Conflict of Interest statement according to the instructions to authors on our website
- include funding information in the Acknowledgement paragraph
- include callouts for figures 4B+C and 7A+B
- make sure that EV2A+B are called out before EV2C
- provide callouts for Appendix Fig S1C, S2A, S3A (which are currently only called out only in the figure legends).
- most appendix figure callouts need the word 'Appendix' adding.
- add page numbers to the table of contents in appendix file 1
- each movie needs to be ZIPed with its legend and the legends removed from the manuscript file.
- check whether figure 1e requires a scale bar
- add scale bars to Appendix Figure S2 panels D&E

The Materials & Methods section contains three tables. Please upload as tables and move them to after the figure legends.

We encourage the publication of source data, particularly for electrophoretic gels and blots, with the aim of making primary data more accessible and transparent to the reader. It would be great if you could provide me with a PDF file per figure that contains the original, uncropped and unprocessed scans of all or key gels used in the figures. The PDF files should be labeled with the appropriate figure/panel number, and should have molecular weight markers; further annotation could be useful but is not essential. The PDF files will be published online with the article as supplementary "Source Data" files. Source Data can also include Excel tables to accompany your graphs. We anticipate that their inclusion will make your work more discoverable and useable to scientists in the future.

We include a synopsis of the paper (see <http://emboj.embopress.org/>). Please provide me with a general summary statement and 3-5 bullet points that capture the key findings of the paper. We also need a summary figure for the synopsis. The size should be 550 wide by [200-400] high (pixels). You can also use something from the figures if that is easier. EMBO Press is an editorially independent publishing platform for the development of EMBO scientific publications.

Best wishes,

William

William Teale, PhD
Editor
The EMBO Journal
w.teale@embojournal.org

We realize that it is difficult to revise to a specific deadline. In the interest of protecting the conceptual advance provided by the work, we recommend a revision within 3 months (6th Oct 2022). Please discuss the revision progress ahead of this time with the editor if you require more time to complete the revisions. Use the link below to submit your revision:

Referee #1:

The authors have now successfully addressed all the suggestions and concerns that I had exposed in my previous review. So, In my opinion the manuscript is now ready to be published in EMBO.J.

Referee #2:

The revised version of the MS adequately addresses the three major points this referee raised in the initial review. The authors are to be praised for their positive response to the reviewer's comments, and essential work has been invested in addressing the collective remarks. The authors have met every reasonable doubt in the revision process. The original submission was judged to be of high quality, substantial novelty, and potentially high impact. The revised MS only strengthens that view. This final version is a significant body of work.

Changes from the last submission:

- 1) Changes on citations to figures according to the editor's requests in the main manuscript, specifically:
 - Updated the conflict of interest statement
 - Included callouts for all figures
 - Changed the title of Figure 7 to "Summary of Findings"
 - Added "Appendix" to appendix figure callouts
 - Added scale bar to Figure 1e
 - Swapped the previous Figure EV2A,B and C,D, in order to call out A,B before C,D;
 - Each movie (.mp4) is now zipped with its legends (.doc) into a zip file, named as "EMBOJ-2022-111021_MovieEVX.zip"
 - Moved the tables to after the main figure legends and named as Table 1-3.
- 2) Changes to the Appendix:
 - We added page numbers to the table of contents.
 - Added scale bars to Appendix Figure S2D,E;
- 3) Source data
 - The source data for each Expanded view figure and Supplementary figure is saved as separate .xlsx files in a folder and the folder is zipped as "EMBOJ-2022-111021_SourceDataForExpandedViewandAppendix.zip"
 - We will also upload additional EV and Appendix image data to the BiImage Archive but, since this is quite a long process, we would prefer to do this later so as not to hold up publication.
- 4) Synopsis
 - We have added a synopsis figure at 550X400 pixel (file name: "EMBOJ-2022-111021_SynopsisImage.jpg") with the summary statement and bullet points (file name: "EMBOJ-2022-111021_Synopsis.docx").
- 5) We have removed the figure titles from the main and Expanded view figures and cropped them to their real sizes.
- 6) We have added E-cadherin signals at the cleavage furrow to the cartoon of the dividing cell in the synopsis of Figure 7. This is to reflect the E-cadherin signals at this region that are in accordance with previous publications.

Dear Clare,

I am pleased to inform you that your manuscript has been accepted for publication in the EMBO Journal.

Congratulations on a wonderful study!

Please note that it is EMBO Journal policy for the transcript of the editorial process (containing referee reports and your response letter) to be published as an online supplement to each paper. If you do NOT want this, you will need to inform the Editorial Office via email immediately. More information is available here:
<https://www.embopress.org/page/journal/14602075/authorguide#transparentprocess>

Your manuscript will be processed for publication in the journal by EMBO Press. Manuscripts in the PDF and electronic editions of The EMBO Journal will be copy edited, and you will be provided with page proofs prior to publication. Please note that supplementary information is not included in the proofs.

You will be contacted by Wiley Author Services to complete licensing and payment information. The required 'Page Charges Authorization Form' is available here: https://www.embopress.org/pb-assets/embo-site/tej_apc.pdf - please download and complete the form and return to embopressproduction@wiley.com

Should you be planning a Press Release on your article, please get in contact with embojournal@wiley.com as early as possible, in order to coordinate publication and release dates.

If you have any questions, please do not hesitate to call or email the Editorial Office. Thank you for your contribution to The EMBO Journal.

Yours sincerely,

William

William Teale, PhD
Editor
The EMBO Journal
w.teale@embojournal.org
